# RL4RLA: Teaching ML to Discover Randomized Linear Algebra Algorithms Through Curriculum Design and Graph-Based Search

**Jinglong Xiong** [* 1]  **Xiaotian Liu** [* 2]  **Ruoxin Wang** [1]  **Zihang Liu** [2]  **Yefan Zhou** [2]  **Yujun Yan** [2]  **Yaoqing Yang** [2]

## Abstract

Randomized linear algebra (RLA) algorithms are a modern class of numerical linear algebra techniques that play an essential role in scientific computing and machine learning, with broad and growing adoption. However, their discovery remains mostly a manual process that requires deep expert knowledge and inspiration. While Reinforcement Learning (RL) offers a pathway to automation, standard approaches struggle with sparse reward landscapes and vast search spaces inherent to high-performing RLA algorithms. In this paper, we present RL4RLA, a general RL framework that automates the discovery of interpretable, symbolic RLA algorithms. Unlike black-box approaches, our method builds explicit algorithms from basic linear algebra primitives, ensuring verifiable and implementable representations. To enable efficient discovery, we introduce: (1) a numerical curriculum that progressively increments problem difficulty to encode inductive bias specific to the RLA domain; (2) Monte Carlo Graph Search, which optimizes exploration by identifying and merging equivalent partial algorithms. We demonstrate that RL4RLA rediscovers state-of-the-art methods, including sketch-and-precondition solvers, Randomized Kaczmarz, and Newton Sketch, and can be targeted to produce algorithms optimized for specific trade-offs between accuracy, speed, and stability. Code is available at https://github.com/Tim-Xiong/RL4RLA.

## 1. Introduction

The idea of learning algorithms directly from data and problem instances has gained attention in recent studies. Notable successes include LION (Chen et al., 2023), which utilizes evolutionary search to discover novel optimization algorithms; AlphaTensor (Fawzi et al., 2022), which decomposes the computation tensor of matrix multiplication to discover new algorithms that outperform Strassen (Strassen, 1969); and AlphaDev (Mankowitz et al., 2023), which uncovered novel, optimized sorting algorithms in assembly code. More recently, LLM-driven approaches such as FunSearch (Romera-Paredes et al., 2024), AlphaEvolve (Novikov et al., 2025), and AlgoTune (Press et al., 2025) have demonstrated that pairing large language models with iterative evaluators can surpass human baselines in mathematics, matrix multiplication, and numerical programming. However, these methods bias search toward the training distribution of the underlying language model, risk missing compositions that fall outside common code patterns, and typically optimize existing implementations rather than composing new algorithmic structures from scratch.

In comparison, randomized linear algebra (RLA) – a cornerstone of large-scale scientific computing and machine learning – has seen far fewer applications of automated algorithm design. RLA algorithms, such as randomized sketching, leverage-score sampling, and stochastic iterative solvers, have been powering systems ranging from large-scale regression to fast low-rank approximations (Halko et al., 2011; Clarkson & Woodruff, 2017). While some previous work addresses learned sketching (Li et al., 2021; Indyk et al., 2019), learned randomized preconditioners (Li et al., 2023; Häusner et al., 2023), and data-driven iterative solvers (Kaneda et al., 2023; Tong et al., 2023), these remain isolated efforts compared to the broad push in other algorithmic domains. To the best of our knowledge, no general-purpose framework currently exists for discovering RLA algorithms to bridge this gap.

RLA exhibits structural properties that make it a promising domain for automated algorithm discovery, particularly its compositional structure. Many sophisticated RLA algorithms arise from compositions of a small set of primitive operators. For instance, randomized SVD decom-

---

[*]Equal contribution  [1]Pratt School of Engineering, Duke University, Durham, NC, USA [2]Department of Computer Science, Dartmouth College, Hanover, NH, USA. Correspondence to: Yaoqing Yang <Yaoqing.Yang@dartmouth.edu>.

*Proceedings of the $43^{rd}$ International Conference on Machine Learning*, Seoul, South Korea. PMLR 306, 2026. Copyright 2026 by the author(s).

poses into sketching, orthonormalization, and deterministic SVD (Halko et al., 2011); randomized least-squares solvers compose subspace embeddings, factorization, and iterative refinement (Clarkson & Woodruff, 2017); and preconditioned methods like Blendenpik (Avron et al., 2010) and LSRN (Meng et al., 2012) integrate sketching with Krylov solvers. Common paradigms include *Sketch-and-Solve* (direct approximate solutions), *Sketch-and-Precondition* (sketching with preconditioned iterative refinement), and *Sketch-and-project* (randomized projection methods). This compositional regularity suggests that RLA algorithm discovery can be formulated as learning to compose a grammar of linear-algebra primitives, which is precisely the kind of structure that reinforcement learning (RL) can exploit.

However, exploiting this structure is nontrivial; there are two severe challenges in discovering RLA algorithms using RL: sparse rewards and large search spaces. Without strong inductive biases, exploration becomes prohibitively expensive. The multi-step, domain-specific techniques required for high-performance numerical computing further amplify these challenges. For instance, the Blendenpik algorithm requires composing at least 5–7 sequential operations: applying a sketching matrix, computing QR factorization, constructing a preconditioner, and iteratively refining the solution. Discovering such multi-step compositions creates combinatorial search spaces that are intractable without structured exploration.

In this paper, we address these challenges by introducing a curriculum-based RL framework for discovering increasingly sophisticated RLA algorithms. The key insight is to construct a sequence of problem instances with increasing difficulty, where each step demands a targeted algorithmic advancement – such as moving from fixed-point iterations to preconditioned solvers, from exact decompositions to sketched factorizations, and from uniform to importance sampling – encoding strong domain knowledge into the search process while still allowing novel combinations to emerge. The main contributions of this work are as follows:

- **General RL framework for interpretable RLA discovery.** We formulate RLA algorithm discovery as constructing explicit symbolic programs from linear-algebra primitives, ensuring interpretability and implementability across problem classes via a domain-agnostic search core.
- **Curriculum decomposition of deep algorithmic search.** We introduce a principled curriculum that decomposes deep algorithmic search into shallow stages, progressively introducing new primitives (sketching, preconditioning, importance sampling) as problem difficulty increases.
- **Systematic recovery and generalization across RLA paradigms.** Our framework rediscovers a broad spec-

trum of RLA algorithms with customizable accuracy, speed, and stability trade-offs. With minimal interface changes, it further generalizes to different problem classes, such as PSD eigenvalue problems.

## 2. Related Work

**Automated Algorithmic Discovery.** Early automated algorithm design used evolutionary and program-synthesis methods. AutoML-Zero (Real et al., 2020) demonstrated algorithm discovery from primitive operations via evolutionary search, while follow-ups extended this paradigm to improve classical routines such as matrix multiplication (Fawzi et al., 2022) and sorting (Mankowitz et al., 2023). Dream-Coder (Ellis et al., 2021) learned reusable program libraries, while LION (Chen et al., 2023) used funnel selection to maintain quality. These established that structured search with inductive bias enables better algorithm discovery.

More recently, LLMs have been integrated into algorithm discovery pipelines. FunSearch (Romera-Paredes et al., 2024) and AlphaEvolve (Novikov et al., 2025) paired LLMs with evaluators to surpass human baselines in mathematics and matrix multiplication. AlgoTune (Press et al., 2025) demonstrated that iterative LLM-driven edit-compile-test loops yield measurable speedups in numerical programming. However, these methods bias search toward training distribution and risk missing compositions that fall outside common code patterns. Neural approaches (Andrychowicz et al., 2016; Veličković et al., 2022) similarly produce black-box representations that limit interpretability.

**Randomized Linear Algebra (RLA).** RLA leverages probabilistic techniques to accelerate large-scale matrix computations (Mahoney, 2011; Woodruff et al., 2014). While foundational work established theoretical guarantees of random projections for matrix approximation (Sarlos, 2006; Drineas et al., 2006; Halko et al., 2011), modern RLA algorithms are composed of two modular primitives: *matrix sketching* and *leverage-score sampling*. Matrix sketching compresses a matrix using embeddings (e.g., Gaussian, CountSketch, SRHT) to preserve spectral geometry (Woodruff et al., 2014; Ailon & Chazelle, 2009). Leverage-score sampling weights rows by their importance to the solution, reducing variance over uniform sampling (Drineas & Mahoney, 2016). These primitives enable efficient algorithms including low-rank approximation (Halko et al., 2011), least-squares regression (Clarkson & Woodruff, 2017), and numerical optimization. Sketch-and-solve methods like Blendenpik (Avron et al., 2010) and LSRN (Meng et al., 2012) combine sketching with iterative refinement to accelerate LSQR and conjugate gradients for overdetermined systems. For nonlinear problems, Newton sketch (Pilanci & Wainwright, 2017) applies sketching to Hessian approximation, enabling scalable second-order methods for logistic regression and general-

ized linear models. Other methods include randomized Kaczmarz (Strohmer & Vershynin, 2009) for linear systems and stochastic gradient methods with variance reduction (Needell et al., 2014).

## 3. Preliminaries

In this section, we briefly review two fundamental concepts underlying our approach: sketching algorithms in RLA and Monte Carlo Tree Search (MCTS) for discrete optimization.

### 3.1. Sketching Algorithms

**Problem Setup.** We consider an overdetermined linear least-squares problem:

$$\min_{x \in \mathbb{R}^n} \|Ax - b\|_2, \tag{1}$$

where $A \in \mathbb{R}^{m \times n}$ has full column rank and $m \gg n$. Classical direct methods such as Cholesky or QR factorization require $\mathcal{O}(mn^2)$ time, which becomes prohibitive when the number of rows is large. RLA addresses this challenge with a geometry-preserving compression method, *sketching*.

**Sketching and Subspace Embedding.** The core primitive in RLA is sketching: applying a random matrix $S \in \mathbb{R}^{s \times m}$ with $s \ll m$ to compress data while preserving the geometry of the column space of $A$.

**Definition 3.1** ($\epsilon$-Subspace Embedding). A matrix $S$ is an $\epsilon$-subspace embedding for $A$ if, for all $x \in \mathbb{R}^n$,

$$(1 - \epsilon)\|Ax\|_2^2 \leq \|SAx\|_2^2 \leq (1 + \epsilon)\|Ax\|_2^2. \tag{2}$$

**Sketch-and-Precondition.** For high-precision applications, sketching is used to accelerate iterative solvers through *preconditioning*. Iterative solvers like Conjugate Gradient converge slowly when the condition number $\kappa(A) = \sigma_{\max}(A)/\sigma_{\min}(A)$ is large. Preconditioning constructs an invertible matrix $M$ such that the transformed system

$$AM^{-1}y = b, \qquad x = M^{-1}y, \tag{3}$$

satisfies $\kappa(AM^{-1}) \approx 1$.

The *Sketch-and-Precondition* paradigm constructs $M$ using a sketched version of the matrix $A$. A typical procedure is

1. **Sketch:** Apply a sketching operator $S \in \mathbb{R}^{s \times m}$ to obtain a reduced matrix $A_S = SA \in \mathbb{R}^{s \times n}$, $s \ll m$.
2. **Factorize:** Compute a matrix factorization, such as QR, $A_S = QR$, to extract a well-conditioned basis.
3. **Precondition:** Use the factor $R$ to define a preconditioner by setting $M = R$. Under standard embedding guarantees, the system $AR^{-1}$ is well-conditioned.

This idea underlies classical RLA solvers such as Blendenpik and LSRN.

**Iterative Solver and Composition.** A preconditioner is often incorporated into an iterative update rule such as

$$x_{t+1} = x_t - \eta M^{-1}(M^{-1})^\top A^\top (Ax_t - b), \tag{4}$$

where $\eta$ is a step size.

**Compositional Structure.** RLA algorithms can be viewed as compositions of a sequence of atomic operations. For example, the discovery of an efficient iterative sketch-and-precondition solver corresponds to identifying the right sequence of transformations that (i) apply a randomized sketch $A \mapsto SA$, (ii) extract a compact representation of the sketched system through a suitable factorization $SA \mapsto R$, and (iii) combine this with an iterative update rule of the form $x_{t+1} \leftarrow \text{Iterate}(A, b, R, x_t)$. This compositional perspective motivates our search-based approach to algorithm discovery in Section 4.

### 3.2. Monte Carlo Tree Search

We formulate algorithm discovery as a sequential decision process over a discrete program space. Each state $s$ represents a partial executable algorithm, and each action $a \in \mathcal{A}(s)$ applies a grammar-defined operation (e.g., inserting an operator at a specific position). Terminal states correspond to complete, executable algorithms.

**MCTS Procedure.** MCTS builds a search tree by iterating four steps: selection, expansion, rollout, and backpropagation (Chaslot, 2010). Actions are chosen to balance exploitation and exploration via an upper confidence bound (UCT) criterion,

$$a' = \arg \max_{a \in \mathcal{A}(s)} \left[ \hat{Q}(s, a) + c \sqrt{\frac{\log N(s)}{N(s, a)}} \right], \tag{5}$$

where $\hat{Q}(s, a)$ is the empirical mean return, $N(s)$ is the visit count of state $s$, $N(s, a)$ is the action visit count, and $c$ is an exploration hyperparameter. A rollout policy completes the partial algorithm to a terminal program within a fixed depth horizon. The resulting candidate algorithm is executed and assigned a scalar reward $R$, reflecting numerical accuracy, stability, and computational cost (Section 4.4). The reward is then backpropagated along the visited path, with corresponding updates to $\hat{Q}(s, a)$, $N(s)$ and $N(s, a)$.

**Challenges in Algorithmic Search** Applying MCTS to algorithmic spaces poses two challenges: (i) *delayed and expensive rewards*, since each rollout requires executing a candidate numerical algorithm; and (ii) *combinatorial explosion of partial programs*, as the branching factor induced by rich operator grammars grows rapidly with program length. These challenges are further amplified in RLA, where effective solvers require composing multiple interacting primitives such as sketching, preconditioning, and sampling.

# 4. Methodology

In this section, we formulate RLA algorithm discovery as constructing explicit symbolic programs from linear algebra primitives (Section 4.1). To address the sparse rewards and vast search spaces challenges, we first introduce curriculum design over problem instances that decomposed deep algorithmic search into a sequence of shallow refinements (Section 4.2); then we use Monte Carlo Graph Search (MCGS), a graph-based generalization of MCTS that merges equivalent partial programs to eliminate redundant exploration (Section 4.3); finally we design reward functions and early stopping mechanisms that allow the framework to discover a wide spectrum of RLA algorithms (Section 4.4).

## 4.1. Algorithm Representation and Search Space

We represent each candidate RLA algorithm as an explicit symbolic program $\mathcal{A} = (\mathcal{P}_{\text{setup}}, \mathcal{P}_{\text{iteration}})$, where $\mathcal{P}_{\text{setup}}$ performs one-time preprocessing (e.g., sketching, factorization) and $\mathcal{P}_{\text{iteration}}$ defines the iterative update rule. This two-stage structure mirrors the canonical organization of modern RLA methods (sketch-factorize-precondition-iterate) and supports curriculum-guided reachability of progressively more sophisticated algorithms.

Programs are constructed incrementally from a library of linear-algebra primitives (e.g., SKETCH, HHQR, MATVEC, INV) operating on typed registers. At each step, the RL agent inserts a single operation target ← operator(operand₁, operand₂) at a chosen position. A type system and legality constraints ensure all generated programs are executable by construction. After each insertion, dead-code elimination removes redundant operations.

As an example, the sketch-and-precondition solver discovered by our framework is:

$$
\begin{aligned}
\mathcal{P}_{\text{setup}}: \quad & R_1 \leftarrow \text{SKETCH}(A), & \mathcal{P}_{\text{iteration}}: \quad & v_1 \leftarrow A x_t, \\
& R_1 \leftarrow \text{QR}(R_1 A), & & v_1 \leftarrow v_1 - b, \\
& R_1 \leftarrow \text{INV}(R_1), & & v_1 \leftarrow A^\top v_1, \\
& R_1 \leftarrow R_1 R_1^\top; & & v_1 \leftarrow R_1 v_1.
\end{aligned}
$$

followed by the update $x_{t+1} = x_t - \eta v_1$. Programs are directly executable. The search procedure, curriculum logic, and MCGS structure are domain-agnostic; only a thin task interface–the operator library, legality constraints, and reward definition (Section 5.4)–varies across problem classes.

## 4.2. Curriculum-Guided Discovery

We introduce a curriculum formulation for algorithm discovery in which the search process is organized as a sequence of staged refinements. Each curriculum stage is associated with a single algorithm, and advancing to the next stage requires augmenting the current algorithm with a single new component. This construction enforces a controlled expansion of the search space.

**Definition 4.1** (Curriculum). A curriculum is defined as an ordered sequence of $S$ stages, denoted by $(\mathcal{C}_s)_{s=1}^{S}$, where each stage is a tuple $\mathcal{C}_s = (A_s, b_s, \mathbf{w}_s)$, comprising family of linear systems parameterized by $(A_s, b_s)$ and a vector of reward weights $\mathbf{w}_s$. During search, each program evaluation samples a fresh random instance from this family, ensuring discovered algorithms generalize across it rather than overfit to a single matrix.

The weights $\mathbf{w}_s$ are designed to emphasize the dominant failure mode of the algorithm from the previous stage. Consequently, the search at stage $s$ initializes from the best algorithm found at stage $s-1$, enabling compositional refinement rather than rediscovery from scratch. We construct $(A_s, b_s)$ with controlled numerical properties via $A = U\Sigma V^\top$: the spectrum $\Sigma$ determines conditioning $\kappa(A) = \sigma_{\max}/\sigma_{\min}$, while the distribution of $U$ controls leverage structure (heavy-tailed $U$ induces high-variance leverage scores). This construction allows us to introduce targeted challenges such as rectangularity, ill conditioning, and stage-wise leverage variability to motivate the evolution of RLA algorithms.

**Example: Sketch-and-Precondition curriculum.** Table 1 illustrates how curriculum design enables compositional discovery through staged failure-mode introduction. We start with a simple $5 \times 5$ well-conditioned system where fixed-point iteration $\boldsymbol{x}_{t+1} = \boldsymbol{x}_t - \eta(\mathbf{A}\boldsymbol{x}_t - \boldsymbol{b})$ already converges reliably, serving as the algorithmic backbone.

We then progressively increase problem difficulty, introducing a failure mode that necessitates a new component. Making the system overdetermined ($m \gg n$) prompts gradient descent via the normal equations. Increasing size to $10000 \times 50$ and introducing ill-conditioning (via controlled spectrum $\mathbf{\Lambda}$ in $\mathbf{A} = \mathbf{U}\mathbf{\Lambda}\mathbf{V}$) causes unpreconditioned gradient descent to converge too slowly, prompting a preprocessing stage that computes $\mathbf{A} = \mathbf{Q}\mathbf{R}^{-1}$ and uses $\mathbf{R}\mathbf{R}^\top$ as a preconditioner. Raising the complexity penalty makes full QR factorization too expensive, prompting sketched preconditioning $\mathbf{S}\mathbf{A} = \mathbf{Q}\mathbf{R}^{-1}$, recovering the main idea of Blendenpik (Avron et al., 2010). Maintaining the same setup allows discovery that stochasticity can further improve performance, initially with uniform row sampling $\mathbf{S}_t$. Finally, sampling $\mathbf{U}$ from a heavy-tailed distribution induces highly non-uniform leverage structure, under which the algorithm learns to sample rows proportional to $\ell_2$ norms.

This decomposition transforms the search for a 6-component algorithm into 5 shallow transitions. Each of them introduces one failure mode resolvable by one new component, keeping the reward landscape locally informative and the search horizon tractable.

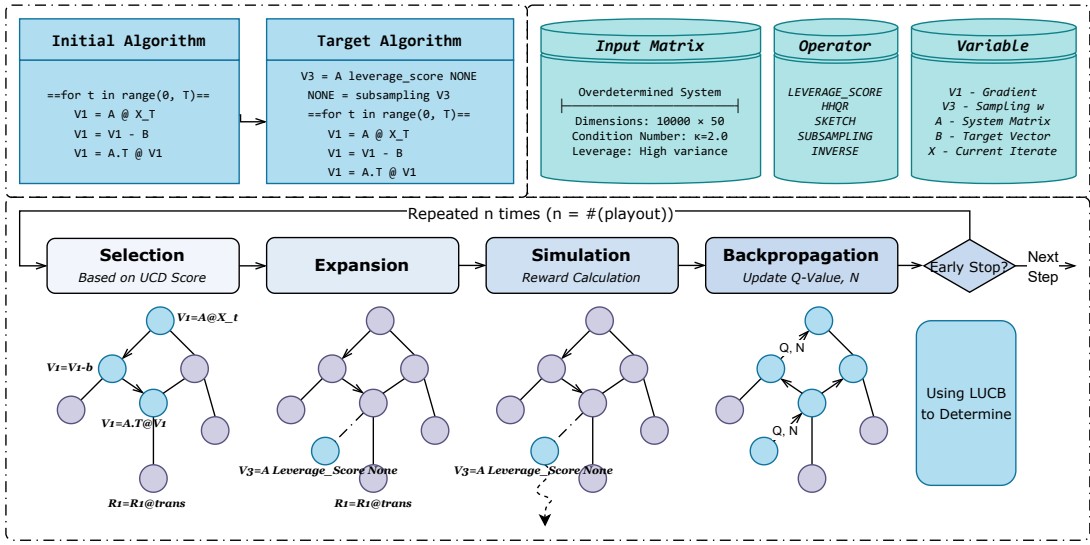

*Figure 1.* Overview of the RL4RLA framework, illustrated with the curriculum transition from Least Square Gradient Descent to Subsampled Least Square Gradient Descent. **Top:** At each curriculum stage, the search is initialized with a base algorithm and a problem environment, including matrix properties, an operator set, and typed variables. The target algorithm is defined by augmenting the current algorithm, as illustrated here by subsampling with leverage-score weighting. **Bottom:** Algorithm discovery is performed by Monte Carlo Graph Search. Each iteration selects actions via UCD, expands the partial algorithm by inserting operators, evaluates candidate algorithms through simulation reward computation, and backpropagates value estimates to update Q-values and visit counts. After each iteration, a LUCB-based criterion determines whether the current curriculum stage has been successfully completed; if so, the discovered partial algorithm is promoted to the next stage, otherwise search continues until the playout budget is exhausted.

### 4.3. Monte Carlo Graph Search

We formulate algorithm discovery as a sequential decision process over partially constructed algorithms, where actions insert operations at specific positions. Standard MCTS treats search space as a tree, causing redundant exploration of equivalent partial algorithms that arise from different action orderings. This redundancy is particularly costly in algorithm discovery, where expensive evaluation and large operator branching factors cause exponential cost growth.

We adopt Monte Carlo Graph Search (MCGS) (Leurent & Maillard, 2020), which operates on a directed acyclic graph (DAG) $\mathcal{G} = (\mathcal{S}, \mathcal{E})$ that merges equivalent states reached via different action sequences. If a state exists, we add an edge to the existing node instead of creating a duplicate. When a rollout from state $s$ yields reward $R$, backpropagation updates statistics along all paths to $s$.

$$N(s, a) \leftarrow N(s, a) + 1, \qquad (6)$$

$$\hat{Q}(s, a) \leftarrow \hat{Q}(s, a) + \frac{R - \hat{Q}(s, a)}{N(s, a)}, \qquad (7)$$

enabling experience from one path to immediately inform decisions along others. States are merged by exact matching of normalized programs (via dead-code elimination), reducing growth from $O(b^d)$ to $O(|\mathcal{S}|)$ unique states.

Concretely, each node in the search graph represent a unique algorithm state. When expanding a node, if the resulting algorithm state already exists in the graph, we reuse the existing node and record an additional parent edge. This design substantially reduces redundant exploration caused by distinct exploration order but semantically equivalent action sequences.

For action selection in the DAG, we use UCD rule (Saffidine et al., 2012):

$$a' = \arg \max_{a \in \mathcal{A}(s)} \left[ \hat{Q}(s, a) + c \sqrt{\frac{\log N(s)}{N(s')}} \right], \qquad (8)$$

where $s'$ is the child state reached by action $a$. This shares exploration statistics across merged states and avoids redundant exploration.

### 4.4. Evaluation Protocol

**Adaptive early stopping.** Manually specified fixed playout budgets introduce human bias and does not provide a principled basis for comparing different search methods. We adopt an adaptive stopping rule based on the Lower and Upper Confidence Bound (LUCB) (Kalyanakrishnan et al., 2012). At each decision step, we identify a *leader* action $a_{\text{leader}} = \arg \max_a \hat{Q}(a)$ and a *challenger* $a_{\text{challenger}} = \arg \max_{a \neq a_{\text{leader}}} [\hat{Q}(a) + U(a)]$, where $\hat{Q}(a)$ is the empirical value and $U(a)$ is the exploration bonus.

*Table 1.* Curriculum progression for sketch-and-precondition discovery. Each stage introduces a single failure mode that is resolved by adding one new algorithmic component.

| Stage | Problem Setup | Failure Mode | Discovered Algorithm | Algorithm Name |
|---|---|---|---|---|
| 0 | $5 \times 5$, well-conditioned | None | $\boldsymbol{x}_{t+1} = \boldsymbol{x}_t - \eta(\mathbf{A}\boldsymbol{x}_t - \boldsymbol{b})$ | Landweber iteration |
| 1 | $m \times n$, rectangular | Non-square system | $\boldsymbol{x}_{t+1} = \boldsymbol{x}_t - \eta\mathbf{A}^\top(\mathbf{A}\boldsymbol{x}_t - \boldsymbol{b})$ | Gradient Descent (GD) |
| 2 | $10000 \times 50$, ill-conditioned | Slow convergence | $\boldsymbol{x}_{t+1} = \boldsymbol{x}_t - \eta\mathbf{R}\mathbf{R}^\top\mathbf{A}^\top(\mathbf{A}\boldsymbol{x}_t - \boldsymbol{b})$ 
 (Preconditioning via $\mathbf{A} = \mathbf{QR}^{-1}$) | Preconditioned GD |
| 3 | Same, higher complexity penalty | High preconditioning cost | $\boldsymbol{x}_{t+1} = \boldsymbol{x}_t - \eta\mathbf{R}\mathbf{R}^\top\mathbf{A}^\top(\mathbf{A}\boldsymbol{x}_t - \boldsymbol{b})$ 
 (Preconditioning via sketch $\mathbf{SA} = \mathbf{QR}^{-1}$) | Sketched Preconditioned GD |
| 4 | Same | Batch gradient inefficiency | $\boldsymbol{x}_{t+1} = \boldsymbol{x}_t - \eta\mathbf{R}\mathbf{R}^\top(\mathbf{S}_t\mathbf{A})^\top(\mathbf{S}_t\mathbf{A}\boldsymbol{x}_t - \mathbf{S}_t\boldsymbol{b})$ 
 ($\mathbf{S}_t$: uniform row sampling) | Uniform Subsampling |
| 5 | Heavy-tailed $\mathbf{U}$ in $\mathbf{A} = \mathbf{U}\boldsymbol{\Lambda}\mathbf{V}^\top$ | Uniform sampling suboptimal | $\boldsymbol{x}_{t+1} = \boldsymbol{x}_t - \eta\mathbf{R}\mathbf{R}^\top(\mathbf{S}_t\mathbf{A})^\top(\mathbf{S}_t\mathbf{A}\boldsymbol{x}_t - \mathbf{S}_t\boldsymbol{b})$ 
 ($\mathbf{S}_t$: $\ell_2$ row-norm sampling) | Leverage-Score Subsampling |

Search terminates when

$$\hat{Q}(a_{\text{leader}}) - U(a_{\text{leader}}) > \hat{Q}(a_{\text{challenger}}) + U(a_{\text{challenger}}), \quad (9)$$

ensuring that sufficient evidence has been accumulated to identify the best action with high confidence. The root advances to the leader's child, and search continues until a maximum depth is reached.

**Reward function.** During search, each candidate algorithm $\mathcal{A}$ is evaluated on the current curriculum stage $(A_s, b_s)$ using a weighted reward

$$R(\mathcal{A}) = \sum_{k \in \mathcal{K}} w_k R_k, \ \mathcal{K} = \{\text{acc}, \text{decay}, \text{comp}, \text{cond}\} \quad (10)$$

where $R_{\text{acc}}$ measures solution accuracy via relative residual $\|Ax - b\|_2/\|b\|_2$, $R_{\text{decay}}$ rewards monotonic convergence by penalizing residual ratio $\rho_{\max} = \max_t \|r_{t+1}\|_2/\|r_t\|_2$, $R_{\text{comp}}$ encourages computational efficiency, and $R_{\text{cond}}$ promotes numerical stability through condition number $\kappa$. The stage-specific weights $\mathbf{w}_s = (w_{\text{acc}}, w_{\text{decay}}, w_{\text{comp}}, w_{\text{cond}})$ emphasize the dominant failure mode at each curriculum stage: early stages prioritize accuracy, while later stages increase complexity penalties to necessitate sketching, or introduce condition rewards to necessitate preconditioning.

## 5. Experiments

We evaluate our framework's ability to discover interpretable and numerically effective randomized linear algebra algorithms. Our experiments examine three key questions: (i) can structured search reliably recover known algorithms across curriculum stages (5.2), (ii) does graph-based search improve exploration efficiency (5.2), and (iii) does the framework generalize beyond least-squares settings and outperform program-search baselines (5.4).

### 5.1. Experimental Setup

#### 5.1.1. CURRICULUM OVERVIEW

We instantiate our framework across five problem domains, each following the same MCGS structure (Section 4.3) but with domain-specific matrix generation, operator spaces, and reward functions.

**RLA Paradigms.** We consider four algorithmic patterns that emerge during curriculum-based discovery: (i) **Sketch-and-Precondition** constructs a randomized preconditioner: given $Ax = b$, a sketching operator $S \in \mathbb{R}^{s \times m}$ yields $SA = QR^{-1}$ such that $\kappa(AR) = \mathcal{O}(1)$, enabling preconditioned gradient updates of the form $x_{t+1} = x_t - \eta R^\top A^\top(Ax_t - b)$. (ii) **Sketch-and-Project** updates iterates by projecting onto randomly sketched subsystems, $x_{t+1} = \arg\min_x \|x - x_t\|_2^2$ s.t. $S_t Ax = S_t b$, recovering methods such as randomized Kaczmarz. (iii) **Sketch-and-Solve** directly solves a sketched least-squares problem $\tilde{x} = \arg\min_x \|SAx - Sb\|_2^2$, and is rediscovered on large overdetermined systems when edit distance is small, reducing complexity while preserving accuracy. (iv) **Newton Sketch** approximates the Hessian $H_t = A^\top D_t A$ as $\hat{H}_t = (SD_t^{1/2}A)^\top(SD_t^{1/2}A)$, reducing complexity from $O(mn^2)$ to $O(smn)$ while maintaining convergence.

#### 5.1.2. EXPERIMENT DETAILS

Each curriculum stage runs MCGS with a fixed simulation budget $B$ and terminates when either the LUCB stopping criterion is satisfied or the budget is exhausted. We define a *discovery event* as successful recovery of a target algorithm within budget, where successes are verified through automated program comparison and manual validation of semantic equivalence.

We evaluate search efficiency using three metrics: (i) **success rate**, the fraction of runs discovering the target algorithm within budget; (ii) **average playouts-to-success**, defined as $\bar{\tau} = \frac{1}{N} \sum_{i=1}^{N} \tau_i$, where $\tau_i$ is the number of playouts

*Table 2.* **End-to-end curriculum completion efficiency across linear-system curricula.** Total playouts and wall-clock time required to reach the final target algorithm across all curriculum transitions (20 runs per transition; LUCB early stopping). SR denotes the success rate of completing the full curriculum within the search budget.

| Target algorithm | Method | Playouts ↓ | Time (s) ↓ / SR |
|---|---|---|---|
| Preconditioned Weighted SGD | MCTS | 34902 | 380.7 / 75% |
| | MCGS+UCT | 13037 | 193.2 / 80% |
| | MCGS+UCD | **10721** | **191.1** / 80% |
| Block Randomized Kaczmarz | MCTS | 66468 | 468.0 / 75% |
| | MCGS+UCT | 38469 | 307.8 / 80% |
| | MCGS+UCD | **25158** | **205.0** / 75% |
| Subsampled Least Square-Gradient Descent | MCTS | 15847 | 10.4 / 75% |
| | MCGS+UCT | 7230 | **8.3** / 80% |
| | MCGS+UCD | **5061** | 8.9 / 80% |
| Sketched Preconditioned Gradient Descent | MCTS | 17655 | 142.9 / 75% |
| | MCGS+UCT | 8030 | **54.8** / 80% |
| | MCGS+UCD | **6034** | 58.4 / 75% |
| Newton Sketch | MCTS | 2557 | 5949.6 / 100% |
| | MCGS+UCT | 2682 | 5265.4 / 100% |
| | MCGS+UCD | **1416** | **4480.9** / 100% |

† Full four-stage curriculum required; all partial-curriculum variants achieve 0% success (Table 7).

in run $i$ and $\tau_i = \infty$ if the target is not discovered within budget; and (iii) **cumulative playouts**, the total search cost from the initial curriculum stage to the current stage. We also track the number of unique states $|S|$ and total node visits $|V|$ to compute the revisit rate $1 - |S|/|V|$.

### 5.2. Discovery Success and Search Efficiency

**Discovery success.** We evaluate five curricula spanning compositional structure, following the algorithmic trajectories illustrated in the curriculum tree (Figure 6). Four curricula operate on linear systems, progressing from a simple base method (Landweber Iteration or Least Square Gradient Descent) to more structured targets (e.g., preconditioned, sketched, subsampled, or block-projection variants); the fifth targets Newton Sketch on logistic regression instances.

For the four linear-system curricula, Figures 2 plot an auxiliary weighted reward (defined only for linear systems and used solely for visualization) against cumulative expected playouts-to-success accumulated from curriculum start. All curves exhibit a consistent staircase pattern: extended plateaus followed by sharp increases aligned with curriculum transitions. These jumps occur because newly introduced structural primitives (e.g., preconditioning, sketching, subsampling, or weighted sampling) enable algorithms to achieve higher accuracy and efficiency across a broader range of linear systems with various numerical properties, while plateaus reflect refinement around the current stage target and its close variants.

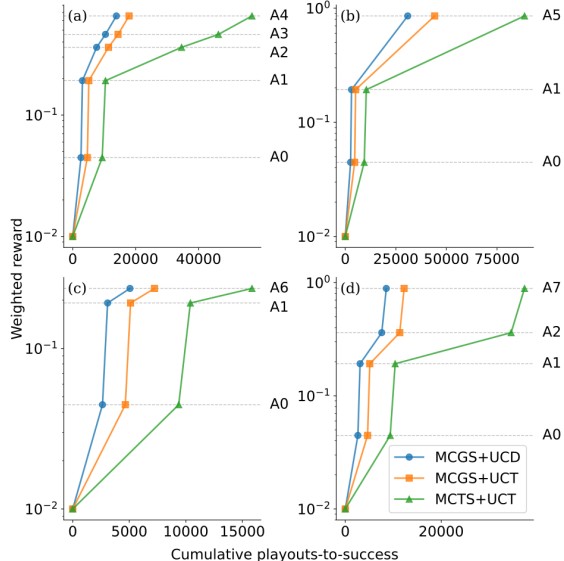

*Figure 2.* **Discovery success across curriculum stages.** Auxiliary weighted reward (linear-systems-only; visualization metric) versus cumulative expected playouts-to-success across curriculum stages. Curves compare MCTS, MCGS with UCT, and MCGS with UCD. Staircase transitions correspond to curriculum stage changes. Four curricula are shown: (a) Preconditioned Weighted Stochastic Gradient Descent (GD), (b) Block Randomized Kaczmarz, (c) Subsampled Least Square GD, and (d) Sketched Preconditioned GD. Stage labels indicate progressively harder algorithmic regimes, from basic Landweber iteration to sketched and subsampled preconditioned methods.

End-to-end curriculum completion (Table 2) shows consistently high success rates (approximately 75-80%) for completing the full curriculum within budget across all linear-system families. The Newton Sketch curriculum further confirms this pattern: the full four-stage curriculum achieves 100% success, while all no-curriculum and partial-curriculum variants fail completely (0%), as analyzed in the ablation study below.

**Search efficiency.** For each run $i$, let $\tau_i$ denote the number of playouts required to reach the target algorithm under LUCB early stopping ($\tau_i = \infty$ if not discovered within budget). We compare methods using Empirical Cumulative Distribution Function (ECDF) $F(x) = (1/N) \sum_i \mathbb{I}(\tau_i \leq x)$, which captures discovery probability at given budgets and sample efficiencies.

Figures 3a-3c report ECDFs for representative targets ranging from shallow (Landweber Iteration) to highly compositional (Sketched Preconditioned GD; Block Randomized Kaczmarz). On shallow targets, all methods achieve comparable discovery speed. As compositional depth increases, both MCGS variants exhibit a pronounced left shift of the discovery-cost distribution relative to MCTS, attaining substantially higher success probability at a fixed budget and

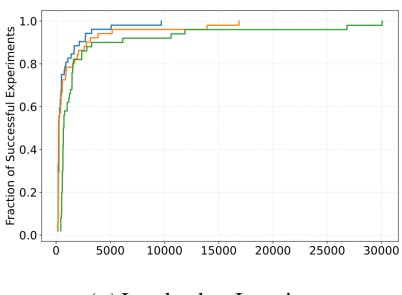
*(a)* Landweber Iteration

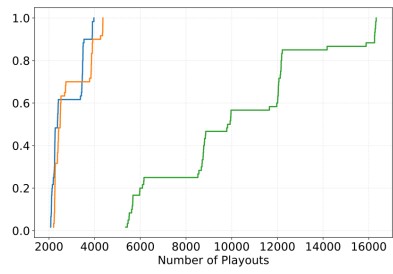
*(b)* Sketched Preconditioned GD

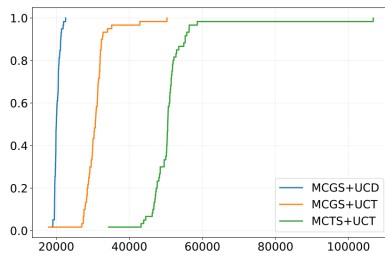
*(c)* Block Randomized Kaczmarz

*Figure 3.* **Search efficiency across representative targets.** Empirical CDF of $\tau$ (playouts required to reach the target under LUCB early stopping), comparing MCTS, MCGS+UCT, and MCGS+UCD. Targets shown: (a) Landweber Iteration, (b) Sketched Preconditioned Gradient Descent (GD), and (c) Block Randomized Kaczmarz. Graph-based search yields consistent sample-efficiency gains, with larger improvements on compositional targets.

reaching high success with markedly fewer playouts.

From Figure 3a to Figure 3c, targets grow in compositional depth. As tasks become more compositional, graph-based search separates from MCTS: MCGS attains substantially higher success probability at a given playout budget. The distinction between MCGS+UCD and MCGS+UCT is modest on simpler targets but becomes most pronounced in the final, most difficult task, where rewards are sparsest. In this regime, UCD avoids over-exploration from multi-parent nodes receiving free exploration bonuses under UCT, leading to clearer gains in success rate and sample efficiency.

To attribute this gap to state reuse, we measure the number of unique states $|S|$ encountered and the revisit rate $1 - |S|/|V|$, where $|V|$ is the total number of visited nodes. Figure 4a shows that MCTS expands nearly linearly in playouts, whereas MCGS grows substantially more slowly, indicating frequent transpositions and effective merging. Figure 4b shows that MCGS sustains a higher revisit rate throughout search, indicating that a larger fraction of compute is spent refining previously encountered partial programs rather than re-generating equivalent ones.

### 5.3. Ablation Studies

**Ablation 1: Curriculum necessity** Stage-wise results (Table 5) show that once a correct base algorithm is provided, subsequent transitions become nearly deterministic (often $\geq$ 95–100% success), indicating that decomposing compositional depth into curriculum stages converts a globally sparse program space into a sequence of locally solvable refinements. This pattern holds across all five curricula and reflects the core mechanism of curriculum design: each stage introduces exactly one failure mode, making the local reward landscape informative and the search horizon tractable. The necessity of curriculum staging is most starkly demonstrated by Newton Sketch. For Newton Sketch, direct end-to-end search and all partial curricula fail completely (0% success), while the full curriculum achieves

100% success, indicating that staged refinement is necessary for reachability rather than efficiency alone. Table 7 details all partial-curriculum variants; in every case, skipping any intermediate stage is sufficient to cause complete failure, ruling out search budget or reward shaping as explanations.

**Ablation 2: Search method** Beyond curriculum design, we also ablate the search procedure itself. Across all curricula, graph-based search (MCGS) reduces total playouts by roughly 2x–3x compared to MCTS. This reduction is driven by state merging: MCGS achieves revisit ratios of 0.50-0.58. The merging benefit degrades as program length and operator library size increase: revisit ratios remain 0.578, 0.530, and 0.520 as target program length grows from 8 to 10 to 12 (library size 17), and decrease modestly from 0.578 to 0.533 as library size grows from 17 to 25 (Table 8), suggesting MCGS remains effective as search complexity grows.

Within MCGS, UCD outperforms UCT on the most compositional targets. UCT assigns free exploration bonuses to multi-parent nodes, causing over-exploration in the DAG setting; UCD corrects this by normalizing against child visit counts. The advantage is modest on simple targets but becomes pronounced when rewards are sparsest. Full ablation results and stage-wise statistics are provided in Appendix D.2.

### 5.4. Generalization and Baseline Comparison

**Generalization to eigenvalue problems.** The experiments above focus on least-squares and linear-system settings, since controlling conditioning and leverage structure isolates the failure modes that drive curriculum construction. To demonstrate transfer to other problem classes, we instantiate RL4RLA on a symmetric PSD eigenvalue problem. The search procedure, curriculum logic, and MCGS structure remain unchanged; the only additions are one normalization primitive (VEC_NORMALIZE) and a Rayleigh quotient-based reward. Across 5 seeds, all three curriculum

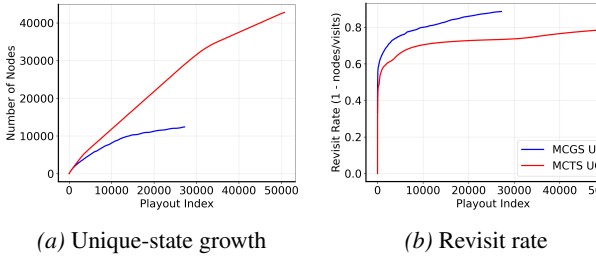

*(a)* Unique-state growth      *(b)* Revisit rate

*Figure 4.* **State reuse under tree vs. graph search (Block Randomized Kaczmarz).** (a) Number of unique algorithmic states $|S|$ versus playout index. (b) Revisit rate $1 - |S|/|V|$ versus playout index, where $|V|$ is the total number of node visits. MCGS saturates earlier than MCTS, consistent with frequent transpositions and node merging. Higher revisit rates under MCGS indicate greater reuse of shared states.

stages succeed, rediscovering power iteration and sketched power iteration at the final stage. This confirms that domain adaptation requires only a thin interface layer – the primitive set and reward definition – while the core framework transfers intact.

*Table 3.* **Curriculum success rates for eigenvalue problems.** Completion success rates across 5 seeds using MCGS+UCD. All three curriculum stages succeed.

| Stage | Size | $\kappa(A)$ | Algorithm | SR |
|-------|------|-------------|-----------|-----|
| C0 | $n = 5$ | $\approx 2$ | Power iter. | 100% |
| C1 | $n = 50$ | $\approx 10^2$ | Power iter. | 100% |
| C2 | $n = 500$ | $\approx 10^2$ | Sketched power iter. | 100% |

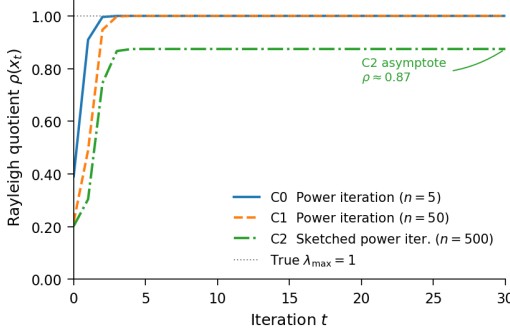

*Figure 5.* **Generalization to eigenvalue problems.** Rayleigh quotient $\rho(x_t)$ over iterations, where $\rho = 1$ indicates exact recovery of $\lambda_{\max}$. C0 and C1 rediscover power iteration ($n = 5$ and $n = 50$), while C2 rediscovers sketched power iteration ($n = 500$). Adapting to this problem class required only one additional primitive, `VEC_NORMALIZE`, and one reward function.

**Comparison with program-search baselines.** RL4RLA differs from LLM-driven program search such as Algo-Tune (Press et al., 2025) and FunSearch (Romera-Paredes et al., 2024): AlgoTune performs code-level optimizations (library swaps, JIT compilation) over existing implemen-

tations rather than composing new algorithmic structures, while FunSearch relies on a pretrained LLM as both mutation operator and inductive bias, biasing search toward the training distribution. In contrast, RL4RLA avoids pretrained priors, searching explicitly over typed symbolic programs. Empirically, FunSearch candidates scale poorly across diverse linear systems, with higher runtime and noncompetitive residuals on both well-conditioned and ill-conditioned systems, while neither baseline discovers interpretable RLA primitives such as sketching or preconditioning. In all five task variants, AlgoTune produces dense deterministic solvers (Cholesky, LAPACK lstsq, or L-BFGS), confirming that code-level optimization does not compose new algorithmic structures. Full comparison tables are provided in Appendix D.4.

## 6. Conclusion and Future Work

We introduced RL4RLA, a curriculum-driven reinforcement learning framework for discovering randomized linear algebra algorithms as explicit symbolic programs. By decomposing deep algorithmic search into staged refinements around numerical failure modes, the framework reliably rediscovers classical RLA paradigms – including sketch-and-precondition, sketch-and-project, and Newton Sketch – within practical search budgets. Equivalent states are merged via Monte Carlo Graph Search, reducing total search cost by 2–3× across all curricula. We also demonstrate that adapting our framework to a new problem class requires only minimal interface changes.

**Limitations.** The discovery environment is synthetic by design, and adapting to a new problem class requires specifying a compact task interface with domain-specific primitives and reward definitions. Moving from rediscovery to validated novel algorithm discovery would further require larger search budgets, richer operator libraries, and systematic post-search formal analysis.

**Future work.** Our work opens several promising directions. First, the operator grammar can be extended to richer iterative solvers and adaptive sketching strategies. Second, learned value or proposal models could be incorporated to further reduce search cost. Finally, building on our eigenvalue results, curriculum-guided discovery can be applied to broader numerical domains such as PDE solvers and large-scale convex optimization.

## Acknowledgements

We thank our colleagues and funding agencies. This work is supported by the DARPA AIQ and DARPA DIAL programs, the U.S. Department of Energy under Award Number DE-SC0025584, and Dartmouth College.

## Impact Statement

This work contributes to automated algorithm discovery in randomized numerical linear algebra. By introducing curriculum-based search over interpretable algorithmic components, our framework can accelerate the exploration of efficient numerical methods for large-scale optimization and scientific computing. This automation may reduce manual design effort and help researchers discover algorithmic variants, particularly when identifying effective compositions of sketching, preconditioning, and sampling strategies.

The discovered algorithms in this work are symbolic programs composed of standard linear-algebra primitives. This enables human inspection, formal verification, and theoretical analysis before deployment. Unlike black-box learned optimizers, our methods produce algorithms that can be understood, adapted, and validated using established numerical analysis techniques.

Users should validate discovered algorithms when deploying in new regimes. We recommend empirical validation on problem instances matching the target application and theoretical analysis of convergence and stability properties when available. The computational cost of search-based discovery should also be considered. In our experiments, curriculum learning enables discovery within practical budgets of hours on standard hardware.

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

## A. Program Space and Legality Constraints

This appendix specifies the formal constraints defining the program space explored by RL4RLA. The goal is to ensure that all candidate programs are executable, type-consistent, and canonical, while avoiding redundancy beyond what is already described in the main text.

**Program form.**   Following Section 4.1, a candidate algorithm is a symbolic program consisting of a preprocessing stage and an iterative stage. Each stage is a sequence of typed linear-algebra operations of the form

$$\texttt{target} \leftarrow \texttt{operator}(\texttt{operand}_1, \texttt{operand}_2),$$

with unary operators using a distinguished `NONE` operand. Programs are constructed incrementally by inserting a single operation at a specified position in either stage. This appendix does not repeat the execution semantics described in the main text, and focuses instead on legality and canonicalization.

**Operator set.**   The search space is defined over a fixed library of 17 linear-algebra operators, summarized in Table 4. Each operator specifies admissible operand types, output type, and allowed execution stage. These operators form a minimal yet expressive basis for composing classical RLA algorithms.

| Category | Operator |
|---|---|
| Vector arithmetic | `VEC_VEC_ADD`, `VEC_VEC_SUB`, `VEC_VEC_DOT` |
| Matrix–vector | `MAT_VEC_MUL`, `VEC_MAT_MUL` |
| Scalar operations | `SCALAR_VEC_MUL`, `SCALAR_DIV` |
| Matrix operations | `MAT_MAT_MUL`, `MAT_MAT_TRANS_MUL`, `MAT_TRANS_MAT_MUL` |
|  | `MAT_INV`, `TRIANGULAR_SOLVE`, `HHQR`, `SKETCH` |
| Sampling and weighting | `SUBSAMPLING`, `LEVERAGE_SCORE` |
| Control | `DO_NOTHING` |

*Table 4.* Operator library defining the program space. All operators enforce type compatibility and stage-specific constraints.

**Legality and redundancy elimination.**   Program legality is enforced through three classes of constraints. First, *type and availability constraints* ensure that operands are defined before use and that all operator inputs and outputs are type-compatible. Read-only system variables are always available but never writable, while cache variables may be written and overwritten subject to availability rules. Second, *grammar constraints* restrict operators to specific execution stages and exclude trivial symmetries, such as operand reordering for commutative operations. Third, *redundancy elimination* removes operations whose outputs are overwritten before being used as operands. This elimination is applied iteratively after each insertion, yielding a canonical representation of each partial program.

A search state is defined by the resulting canonical program. Two states are considered identical if their programs are identical after redundancy elimination, which enables safe state merging in Monte Carlo Graph Search.

## B. Curriculum and Problem Instance Construction

We evaluate discovered algorithms on three classes of synthetic problem instances: linear systems, logistic regression systems, and eigenvalue problems. All are constructed to exhibit controlled numerical properties, enabling systematic isolation of the failure modes that drive curriculum design.

**Linear systems.**   Instances take the form $Ax = b$ and are generated together with a reference solution $x^\star$ satisfying $b = Ax^\star$, which is used only for evaluation. We consider four instance families with increasing numerical difficulty. **PSD** systems use symmetric positive definite matrices with explicitly prescribed condition number. **LOW-COND**, **MID-COND**, and **HIGH-COND** systems are rectangular or square matrices constructed via a singular value decomposition

$$A = U\Sigma V^\top,$$

which allows direct control over the spectrum and conditioning. LOW-COND systems use narrow spectra with small condition number, MID-COND systems use moderately spread spectra, and HIGH-COND systems use rapidly decaying spectra, resulting in severe ill-conditioning.

To control leverage-score variation, the left singular vectors $U$ are sampled either from standard Gaussian ensembles, yielding approximately uniform leverage scores, or from heavy-tailed distributions, yielding highly non-uniform leverage scores. This construction creates regimes where adaptive sampling and preconditioning are either unnecessary, beneficial, or essential. The right-hand side $b$ is generated to lie in the column space of $A$ by default, producing consistent systems and ensuring that differences in performance are driven by numerical conditioning rather than infeasibility.

**Logistic regression systems.**   The Newton Sketch curriculum operates on logistic regression instances rather than linear systems, since Newton Sketch is a second-order method for nonlinear objectives. Each instance defines a binary classification problem with data matrix $X \in \mathbb{R}^{m \times n}$ and labels $y \in \{-1, +1\}^m$, with objective

$$\min_{w \in \mathbb{R}^n} \sum_{i=1}^{m} \log\big(1 + \exp(-y_i x_i^\top w)\big).$$

The data matrix $X$ is generated as $X = U\Sigma V^\top$ with the same SVD-based construction used for linear systems, allowing direct control over conditioning and leverage structure. Labels are generated as $y_i = \text{sign}(x_i^\top w^\star + \epsilon_i)$, where $w^\star$ is a fixed ground-truth weight vector and $\epsilon_i \sim \mathcal{N}(0, \sigma^2)$ controls label noise.

The curriculum spans four stages of increasing difficulty. Stage 1 (Logistic Forward Pass) uses a small well-conditioned instance ($m = 50$, $n = 10$, $\kappa \approx 2$) where computing the sigmoid activation and cross-entropy loss is sufficient. Stage 2 (Logistic GD) increases size ($m = 500$, $n = 20$) and requires a first-order gradient update. Stage 3 (Full Newton) introduces a large ill-conditioned instance ($m = 5000$, $n = 50$, $\kappa \approx 10^3$), where slow first-order convergence necessitates the exact Hessian $H = X^\top D X$ with $D = \text{diag}(\sigma(Xw)(1 - \sigma(Xw)))$. Stage 4 (Newton Sketch) maintains the same instance but introduces a high computational cost penalty, making exact Hessian computation prohibitive and necessitating the sketched approximation $\widehat{H} = (SD^{1/2}X)^\top (SD^{1/2}X)$.

The reward at each stage is based on the cross-entropy loss relative to a baseline first-order solver, combined with a computational cost penalty whose weight increases from stage 3 to stage 4 to expose the cost bottleneck that motivates sketching.

**Eigenvalue problems.**   To evaluate generalization beyond least-squares settings, we construct symmetric PSD eigenvalue instances of the form $Av = \lambda v$, where $A \in \mathbb{R}^{n \times n}$ is symmetric positive definite. Each instance is generated as $A = Q\Lambda Q^\top$, where $Q$ is a random orthogonal matrix drawn from the Haar measure and $\Lambda = \text{diag}(\lambda_1, \ldots, \lambda_n)$ has eigenvalues sampled log-uniformly over a prescribed range $[\lambda_{\min}, \lambda_{\max}]$, controlling the condition number $\kappa(A) = \lambda_{\max}/\lambda_{\min}$. The curriculum spans three stages of increasing difficulty: a well-conditioned instance ($\kappa \approx 2$) at stage 1, a moderately ill-conditioned instance ($\kappa \approx 10^2$) at stage 2, and a large ill-conditioned instance with sketching at stage 3.

The reward at each stage is based on the Rayleigh quotient:

$$R_{\text{eig}}(v) = \frac{v^\top A v}{v^\top v},$$

which measures how well the discovered iterate $v$ approximates the leading eigenvector. The reward is log-scaled and normalized relative to a random-initialization baseline to produce a signal comparable in magnitude to the linear-system reward. No other reward components or curriculum logic change relative to the linear-system setting; the only additions to the operator library are the VEC_NORMALIZE primitive and the Rayleigh quotient reward computation.

## C. Semantic Equivalence and Canonicalization

A discovery event is declared successful only when the discovered program is verified to be semantically equivalent to the target algorithm. Equivalence checking proceeds in two steps.

**Step 1: Symbolic canonicalization.**   Each program is first reduced to a canonical form using a set of algebraic simplification rules. These include collapsing $A^\top A$ products into a single matrix operation, eliminating identity operators, and normalizing scalar multiplications. Canonicalization is fully automated and requires no manual intervention.

**Step 2: Execution-based equivalence testing.**   Two programs whose canonical forms match are further validated by executing both on a held-out set of random matrix instances and comparing outputs. Two programs are declared equivalent

if and only if both symbolic canonicalization and execution-based testing agree.

Manual validation is limited to spot-checking a random sample of declared-equivalent pairs to verify that neither step produces false positives. In practice, no false positives were found across all reported experiments. The canonicalization pipeline will be released as part of our code release to support reproducibility.

## D. Additional Experimental Details

### D.1. Auxiliary Weighted Reward (Reporting Only)

To summarize algorithm performance across linear systems with different numerical characteristics, we report an auxiliary *weighted aggregate reward*. This quantity is used only for reporting and visualization. Importantly, the reward computed on each individual environment follows exactly the same definition as that used during search.

Let $\mathcal{E}$ denote a set of linear-system environments. For a discovered algorithm $\mathcal{A}$ and environment $e \in \mathcal{E}$, we evaluate $\mathcal{A}$ using the reward function defined in Section 4.4, after selecting the best execution over a fixed learning-rate grid

$$\mathcal{L} = \{0.01, 0.03, 0.07, 0.1, 0.3, 0.7, 1.0\}$$

based on the smallest loss. This yields an environment-specific reward value $R(\mathcal{A}; e)$.

An environment profile specifies nonnegative weights $\{\omega_e\}_{e \in \mathcal{E}}$. We normalize these weights and define the auxiliary weighted reward as

$$R_{\text{aux}}(\mathcal{A}) = \sum_{e \in \mathcal{E}} \frac{\omega_e}{\sum_{e' \in \mathcal{E}} \omega_{e'}} R(\mathcal{A}; e).$$

We use two fixed profiles throughout the paper: `ht`, with weights $(1, 5, 10)$ over $\{$`psd`, `ht_low_cond`, `ht_mid_cond`$\}$, and `non_ht`, with weights $(1, 5, 10)$ over $\{$`psd`, `low_cond`, `mid_cond`$\}$. All environment-level rewards use the same component weights as in Section 4.4, $(w_{\text{acc}}, w_{\text{decay}}, w_{\text{comp}}, w_{\text{cond}}) = (5, 1, 8, 0)$.

We emphasize that $R_{\text{aux}}(\mathcal{A})$ is a post-hoc aggregation across environments and does not affect search, curriculum progression, or early stopping, which operate solely on $R(\mathcal{A}; e)$ within a single environment.

### D.2. Curriculum Trajectories and Ablations

Figure 6 visualizes the end-to-end discovery trajectory induced by curriculum-guided search. Rather than discovering target algorithms in isolation, search first converges to a shared trunk of increasingly refined solvers, which is then reused across multiple downstream targets. This shared structure substantially reduces redundant exploration and highlights the compositional nature of the discovered algorithmic space.

Tables 5 and 6 make this effect explicit. Each row corresponds to a curriculum transition and reports the edit distance between source and target programs together with discovery cost and success rate. Transitions with small edit distance are consistently easy to discover, while transitions introducing new numerical mechanisms (e.g., sketching or preconditioning) form clear bottlenecks. Crucially, once such bottlenecks are crossed, subsequent refinements become dramatically cheaper, demonstrating that curriculum design enables search to amortize difficult discoveries across multiple targets.

Ablations on Newton Sketch (Table 7) further establish curriculum necessity. Direct end-to-end search fails completely, as do all partial curricula that skip intermediate stages (e.g., jumping from Logistic Forward Pass directly to Full Newton or Newton Sketch, or starting from Logistic GD), despite identical budgets and reward definitions. Only the full four-stage curriculum—Logistic Forward Pass → Logistic GD → Full Newton → Newton Sketch—achieves consistent success across all seeds. This result rules out search budget or reward shaping as explanations, and instead demonstrates that staged exposure to intermediate failure modes is a prerequisite for discovering deep, compositional algorithms in non-linear settings.

### D.3. MCGS Scalability Analysis

To analyze how the state-merging benefit in MCGS scales with search complexity, we ran MCGS and MCTS on a controlled task (rediscovering Sketched Preconditioned GD, a base program of 8 ops) across varying operator library sizes (17, 20, 25) and target program lengths (edit distance 8, 10, 12), with 20 seeds each. We report the *revisit ratio*: the fraction of search steps redirected to already-visited states (MCTS = 0.000 in all settings).

*Table 5.* **Stage-wise discovery cost breakdown across linear-system curricula.** For each curriculum transition, each cell reports playouts / time (s) / success rate (20 runs; LUCB early stopping). Boldface marks the best playout count and the best wall-clock time per transition.

| Curriculum | Transition | MCTS | MCGS+UCT | MCGS+UCD |
|---|---|---|---|---|
| Preconditioned Weighted SGD | NULL → Landweber Iteration | 9371 / 0.6 / 75% | 4651 / 0.3 / 80% | **2632** / **0.2** / 80% |
| | Landweber Iteration → Least Square Gradient Descent | 1032 / **0.4** / 100% | 464 / **0.4** / 100% | **453** / 0.4 / 100% |
| | Least Square Gradient Descent → Preconditioned Gradient Descent | 4456 / **7.8** / 100% | 1999 / 7.9 / 100% | **1999** / 7.9 / 100% |
| | Preconditioned Gradient Descent → Sketched Preconditioned Gradient Descent | 11691 / 366.3 / 100% | 3044 / 179.7 / 100% | **2800** / **177.4** / 100% |
| | Sketched Preconditioned Gradient Descent → Preconditioned Weighted SGD | 8353 / 5.6 / 100% | 2880 / **4.9** / 100% | **2838** / 5.2 / 100% |
| | **Total playouts** | 34902 | 13037 | **10721** |
| Block Randomized Kaczmarz | NULL → Landweber Iteration | 9371 / 0.6 / 75% | 4651 / 0.3 / 80% | **2632** / **0.2** / 80% |
| | Landweber Iteration → Least Square Gradient Descent | 1032 / **0.4** / 100% | 464 / **0.4** / 100% | **453** / 0.4 / 100% |
| | Least Square Gradient Descent → Block Randomized Kaczmarz | 56066 / 467.0 / 100% | 33354 / 307.1 / 100% | **22074** / **204.4** / 95% |
| | **Total playouts** | 66468 | 38469 | **25158** |
| Subsampled Least Square GD | NULL → Landweber Iteration | 9371 / 0.6 / 75% | 4651 / 0.3 / 80% | **2632** / **0.2** / 80% |
| | Landweber Iteration → Least Square Gradient Descent | 1032 / **0.4** / 100% | 464 / **0.4** / 100% | **453** / 0.4 / 100% |
| | Least Square Gradient Descent → Subsampled Least Square Gradient Descent | 5445 / 9.4 / 100% | 2115 / **7.6** / 100% | **1977** / 8.3 / 100% |
| | **Total playouts** | 15847 | 7230 | **5061** |
| Sketched Preconditioned GD | NULL → Landweber Iteration | 9371 / 0.6 / 75% | 4651 / 0.3 / 80% | **2632** / **0.2** / 80% |
| | Landweber Iteration → Least Square Gradient Descent | 1032 / **0.4** / 100% | 464 / **0.4** / 100% | **453** / 0.4 / 100% |
| | Least Square Gradient Descent → Preconditioned Gradient Descent | 4456 / **7.8** / 100% | 1999 / 7.9 / 100% | **1999** / 7.9 / 100% |
| | Preconditioned Gradient Descent → Sketched Preconditioned Gradient Descent | 2796 / 134.1 / 100% | **916** / **46.2** / 100% | 951 / 49.9 / 95% |
| | **Total playouts** | 17655 | 8030 | **6034** |

*Table 6.* **Curriculum tree (MCGS+UCD).** Each row is a stage transition (edge) in the shared-trunk curriculum. Columns correspond to leaf targets; entries report playouts (median $\pm$ IQR) / success rate over 20 seeds. Dashes indicate the transition is not on the path. **Edit dist.** is the edit distance between the source and target programs for the transition.

| Transition | Edit dist. | Subsampled LS-GD | Sketched Precond GD | Precond Weighted SGD | Block RK |
|---|---|---|---|---|---|
| *Shared trunk* | | | | | |
| NULL → Landweber | 2 | $1238 \pm 2013/80\%$ | $1238 \pm 2013/80\%$ | $1238 \pm 2013/80\%$ | $1238 \pm 2013/80\%$ |
| Landweber → LS-GD | 1 | $451 \pm 9/100\%$ | $451 \pm 9/100\%$ | $451 \pm 9/100\%$ | $451 \pm 9/100\%$ |
| *Branches* | | | | | |
| LS-GD → Subsampled LS-GD | 2 | $1969 \pm 32/100\%$ | — | — | — |
| LS-GD → preconditioner $R_1$ | 3 | — | $1730 \pm 27/70\%$ | $1730 \pm 27/70\%$ | — |
| $R_1$ → Precond GD | 1 | — | $1999 \pm 0/100\%$ | $1999 \pm 0/100\%$ | — |
| Precond GD → Sketched Precond GD | 2 | — | $904 \pm 2/95\%$ | $2390 \pm 1636/100\%$ | — |
| Sketched Precond GD → subsampling | 1 | — | — | $626 \pm 0/100\%$ | — |
| subsampling → Precond Weighted SGD | 1 | — | — | $2836 \pm 543/100\%$ | — |
| LS-GD → sketch & project base | 1 | — | — | — | $5731 \pm 110/100\%$ |
| sketch & project base → Block RK | 3 | — | — | — | $20979 \pm 799/95\%$ |

MCGS consistently achieves revisit ratios of 0.50–0.58, meaning over half of all search steps exploit previously merged states rather than spawning redundant subtrees. Two trends are visible: the revisit ratio decreases modestly as program length grows (0.578→0.520 at library size 17) and as library size grows (0.578→0.533 at length 8), since both increase the structural diversity of partial programs. Importantly, both effects are moderate, so the merging benefit degrades gracefully rather than collapsing, suggesting that MCGS remains effective as search complexity increases.

### D.4. Baseline Comparison

We compare RL4RLA against two LLM-driven program search baselines: AlgoTune (Press et al., 2025) and FunSearch (Romera-Paredes et al., 2024). The key architectural distinction is that AlgoTune performs code-level optimizations (library swaps, JIT compilation) over existing implementations rather than composing new algorithmic structures, while FunSearch relies on a pretrained LLM as both mutation operator and inductive bias. Neither baseline is designed to discover symbolic RLA primitives such as sketching or preconditioning from scratch.

FunSearch candidates scale poorly across diverse linear systems, producing orders-of-magnitude higher runtime and noncompetitive residuals on both well-conditioned and ill-conditioned systems. AlgoTune achieves competitive runtime on the specific system it is tuned for by exploiting library-level optimizations, but does not generalize across matrix families and does not produce reusable algorithmic structure.

*Table 7.* **Curriculum necessity for Newton Sketch.** Success rate for reaching the target stage under different curriculum designs (20 runs; LUCB early stopping). No-curriculum and all partial-curriculum variants fail completely; only the full four-stage curriculum achieves consistent success.

| Curriculum Design | Target Stage | Success Rate |
|---|---|---|
| No curriculum | Newton Sketch | 0% |
| Partial: Logistic Forward Pass → Full Newton | Full Newton | 0% |
| Partial: Logistic Forward Pass → Newton Sketch | Newton Sketch | 0% |
| Partial: Logistic Gradient Descent (GD) → Newton Sketch | Newton Sketch | 0% |
| Full: Logistic Forward Pass → Logistic GD → Full Newton → Newton Sketch | Newton Sketch | 100% |

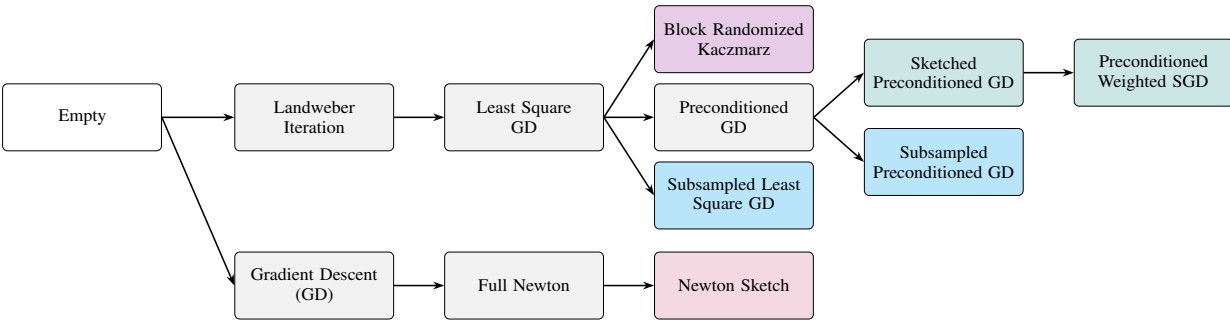

*Figure 6.* Algorithm discovery trajectory across curriculum stages. The search begins from an empty program and progressively builds complexity through operator composition. Four algorithmic paradigms emerge: Sketch-and-Precondition (teal), Sketch-and-Project (purple), Sketch-and-Solve (blue), and Newton Sketch (pink). Intermediate base methods are shown in gray.

## D.5. Evaluation on Real-world Dataset

To complement the controlled discovery setting, we evaluate the algorithms discovered under synthetic curricula on the real-world `YearPredictionMSD` dataset. Figure 8 shows the comparison of final relative residual with respect to runtime on this dataset, and Table 9 reports residual and wall-clock time for four discovered algorithms. The results provide initial evidence that the learned algorithmic structures transfer beyond the synthetic instances used during search; notably, Preconditioned GD and Sketched Preconditioned GD achieve the same residual while the latter incurs moderate additional cost from sketching step.

## E. Benchmark and Dataset Description

### E.1. Synthetic Benchmarks

The primary evaluation setting uses synthetic linear systems, logistic regression instances, and eigenvalue problems constructed with controlled numerical properties, as described in Appendix B. These synthetic benchmarks are the main discovery and evaluation environment by design: controlling the spectrum, conditioning, and leverage structure of instances allows the curriculum to isolate specific failure modes that motivate algorithmic components, and the symbolic program representation makes discovered algorithms directly verifiable against known methods.

### E.2. YearPredictionMSD

To evaluate transfer of discovered algorithms to a real-world setting, we use the **YearPredictionMSD** dataset (Bertin-Mahieux, 2011) from the UCI Machine Learning Repository. The dataset contains 515,345 instances, each describing a song by 90 audio features: 12 timbre averages and 78 timbre covariance features extracted from the Echo Nest API, representing averaged and covariance statistics over all temporal segments of a track. The regression target is the song's release year, ranging from 1922 to 2011, with a peak in the 2000s. The dataset contains no missing values and all features are real-valued.

We follow the official train/test split recommended by the dataset creators: the first 463,715 examples form the training set and the remaining 51,630 form the test set. This split is designed to avoid the *producer effect*, ensuring that no song from a given artist appears in both train and test sets.

*Table 8.* **MCGS revisit ratio across operator library sizes and program lengths.** Higher values indicate greater state reuse. MCTS achieves 0.000 in all settings.

| Library size | Length 8 | Length 10 | Length 12 |
|---|---|---|---|
| 17 | 0.578 | 0.530 | 0.520 |
| 20 | 0.578 | 0.527 | — |
| 25 | 0.533 | 0.499 | — |

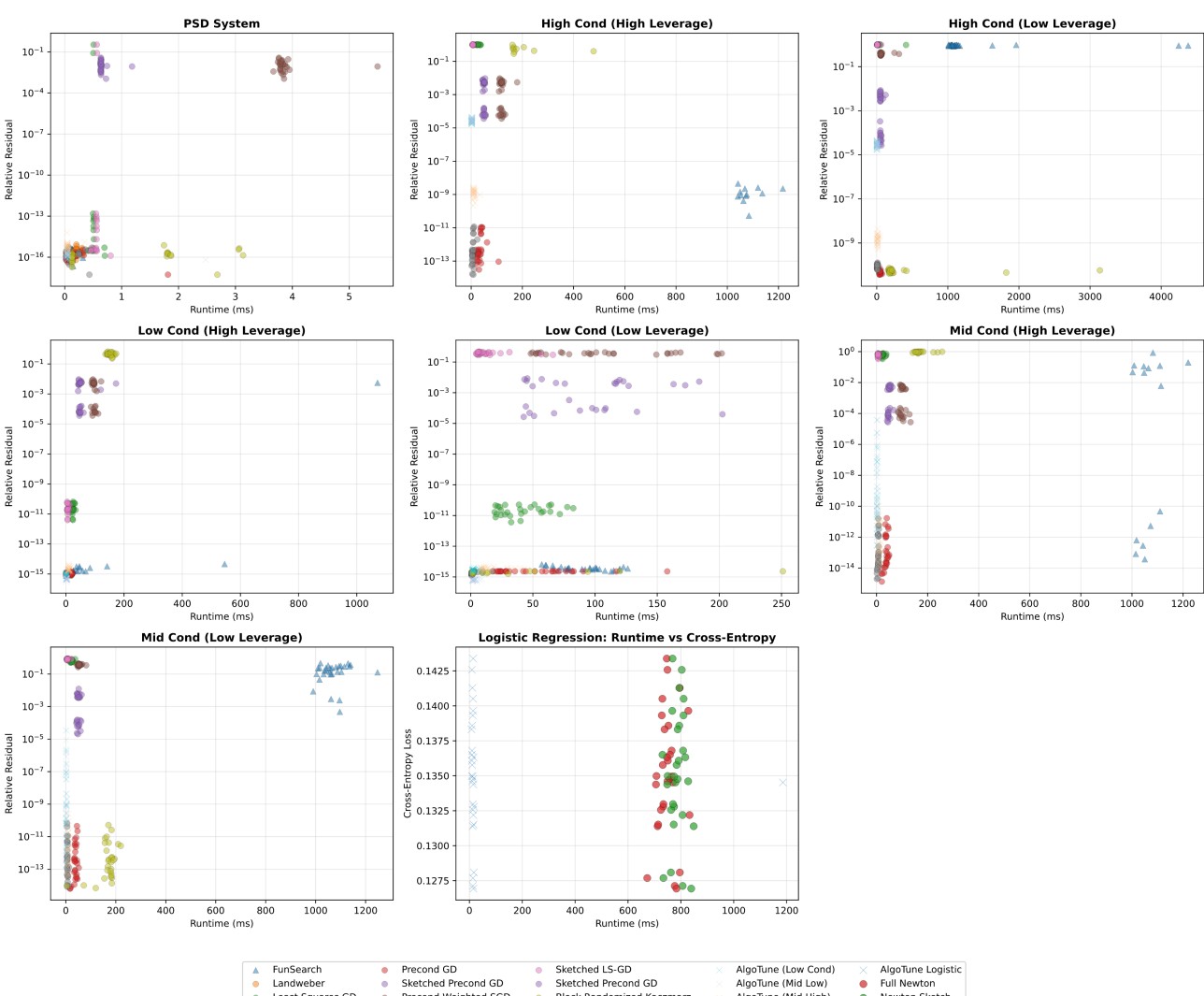

*Figure 7.* Runtime vs. residual for RL4RLA-discovered methods, FunSearch and AlgoTune variants (Mid Low, Mid High, Logistic) across 5 system families. Each subplot represents one system configuration.

**Usage in this work.** We do not use YearPredictionMSD as a training signal for algorithm discovery. The algorithms reported in Section D.5 are discovered entirely on synthetic instances. YearPredictionMSD is used solely for *post-hoc transfer evaluation*: the discovered symbolic programs are executed on the dataset without any modification to reward design or search hyperparameters. This tests whether the algorithmic structures found under controlled synthetic conditions generalize to a large-scale, real-world regression problem with natural leverage and spectral structure.

Concretely, we formulate the dataset as an overdetermined least-squares problem $\min_x \|Ax - b\|_2^2$, where $A \in \mathbb{R}^{463715 \times 90}$ is the training feature matrix and $b \in \mathbb{R}^{463715}$ is the vector of release years. We evaluate residual $\|Ax - b\|_2/\|b\|_2$ and

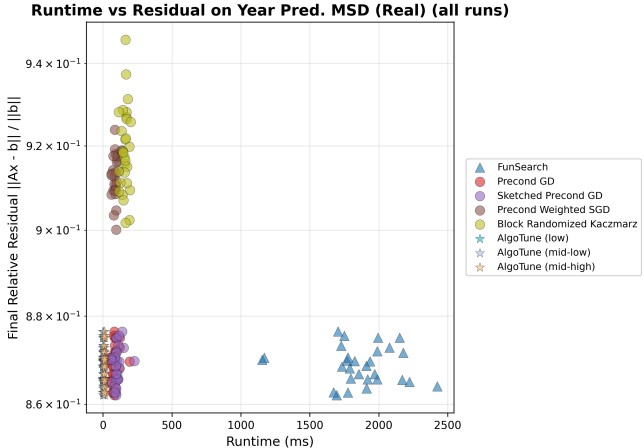

*Figure 8.* Runtime vs. final relative residual on YearPredictionMSD. Each point corresponds to one of 30 independent runs.

*Table 9.* **Real-world transfer on `YearPredictionMSD`.** Discovered algorithms evaluated without retraining or reward changes.

| Algorithm | Residual ↓ | Time (s) ↓ |
|---|---|---|
| Precond GD | 8.69e-01 | 85.64 |
| Sketched Precond GD | 8.69e-01 | 103.46 |
| Precond Weighted SGD | 9.13e-01 | 92.22 |
| Block RK | 9.19e-01 | 159.14 |

wall-clock time on a fixed hardware configuration.

## F. Baseline Implementation Details

We compare RL4RLA against two LLM-driven program search baselines, AlgoTune (Press et al., 2025) and Fun-Search (Romera-Paredes et al., 2024). This appendix describes the setup used for each baseline and clarifies the architectural distinction from RL4RLA.

### F.1. AlgoTune

AlgoTune (Press et al., 2025)is a benchmark and agent system for LLM-driven code optimization. For each task, an LLM-powered agent called *AlgoTuner* iteratively rewrites a reference implementation to maximize runtime speedup while preserving output correctness on a held-out test set.

**Task construction.** We implement a separate custom AlgoTune task class for each RL4RLA curriculum environment, registered via the @register_task decorator, so that AlgoTuner operates on matrix families directly matched to the numerical difficulty of the corresponding curriculum stage. Table 10 summarizes the five task variants and their environment configurations.

*Table 10.* **AlgoTune task variants and environment configurations.** Each task corresponds to one RL4RLA curriculum environment. System type controls conditioning; lev controls leverage-score distribution; vt-dis controls the right singular vector ensemble.

| Task name | RL4RLA target | System type | Rows | Cols | Leverage | $V^\top$ |
|---|---|---|---|---|---|---|
| rl4rla_low | Subsampled LS-GD | LOW_COND | 10000 | 50 | high | Gauss |
| rl4rla_mid_low | Sketched Precond GD | MID_COND | 10000 | 50 | low | Gauss |
| rl4rla_mid_high | Precond Weighted GD / Block RK | MID_COND | 10000 | 50 | high | Gauss |
| rl4rla_brk | Block Randomized Kaczmarz | MID_COND | 10000 | 50 | low | Gauss |
| rl4rla_logistic | Newton Sketch | LOGISTIC_REG | 10000 | 100 | — | — |

For the linear-system tasks, each instance generates $A \in \mathbb{R}^{m \times n}$ via $A = U\Sigma V^\top$, where conditioning, leverage structure,

and $V^\top$ distribution are set according to Table 10. The reference solver is `scipy.linalg.lstsq`. A solution is accepted if $\|Ax - b\|_2/\|b\|_2 < 0.01$. For the logistic task ($m = 10000$, $n = 100$, sep = 5.0), the reference solver runs 50 steps of full Newton's method with exact Hessian $H = A^\top \mathrm{diag}(p \odot (1 - p))A/m + 10^{-8}I$ solved via `scipy.linalg.solve`. A solution is accepted if binary cross-entropy falls below 0.5.

**AlgoTuner solutions.** Table 11 reports the solver discovered by AlgoTuner for each task. In all cases, AlgoTuner produces a *dense deterministic solver*: normal equations with Cholesky factorization (via `numpy.linalg.cholesky` or BLAS `dsyrk` + `cho_factor`), LAPACK `lstsq`, or `sklearn` L-BFGS. None of the discovered solutions incorporate randomized structures such as sketching, leverage-score sampling, or randomized preconditioning.

*Table 11.* **AlgoTuner solutions per task.** All discovered solvers are dense and deterministic.

| Task name | Discovered solver |
|---|---|
| rl4rla_low | Normal equations + Cholesky (`numpy.linalg.cholesky`) |
| rl4rla_mid_low | Normal equations + Cholesky (BLAS `dsyrk` + `cho_factor`) |
| rl4rla_mid_high | LAPACK `scipy.linalg.lstsq` |
| rl4rla_brk | Normal equations + Cholesky (`numpy.linalg.cholesky`) |
| rl4rla_logistic | `sklearn LogisticRegression` (L-BFGS, no penalty) |

**Search setup.** AlgoTuner is given access to `numpy`, `scipy.linalg`, and `sklearn`. The target runtime is 10 ms per instance. We use GPT-4o as the underlying LLM and allow a search budget of 20 iterations per task.

**Key distinction from RL4RLA.** AlgoTuner optimizes *existing code implementations*: it discovers faster ways to call the same class of solver (e.g., exploiting positive definiteness to use Cholesky instead of general LU, or selecting a faster BLAS routine) rather than composing new algorithmic structures from symbolic primitives. As a result, AlgoTuner solutions neither discover randomized algorithms nor generalize across qualitatively different matrix families – an AlgoTuner solution tuned for LOW_COND instances (where Cholesky on the normal equations is stable) is not appropriate for HIGH_COND instances where the normal equations are ill-conditioned and sketching or preconditioning is necessary.

## F.2. FunSearch

We run FunSearch (Romera-Paredes et al., 2024) in the usual formulation: a programs database maintains multiple evolutionary islands, and each island proposes offspring by mutating stored code. In our setup, mutations are produced through the OpenAI API using the model `gpt-5-mini-2025-08-07`. The only code the model may change is the body of a single iterative routine, initialized to a Jacobi splitting (diagonal solve with off-diagonal coupling). An outer evaluation harness that wraps this routine, runs it on the test batch, and returns a scalar score is fixed throughout search. System-level instructions encourage small, local edits rather than wholesale rewrites. Table 12 lists the exact initial routine and prompt strings.

Each experimental run draws one *fixed* batch of four systems – PSD, LOW_COND, MID_COND, and HIGH_COND – using the construction of Appendix B, so the objective does not shift when new candidates are evaluated. Rectangular families use $10{,}000 \times 50$ matrices; PSD uses a small dense $5 \times 5$ system with controlled condition number. Unless otherwise configured, non-PSD instances use high leverage and Student-$t$ right singular vectors, with a shared random seed for reproducibility. Every candidate is executed for at most 10 iterations with tolerance $10^{-6}$. For each system we also form a sparse Johnson–Lindenstrauss sketch so that computational cost is accounted for in the same way as in the RL4RLA executor.

The scalar score matches RL4RLA's multi-term reward: normalized residual accuracy, stability via consecutive residual ratios, complexity from a FLOP count, and optional conditioning terms, combined with log aggregation and component weights $(1, 10, 8, 1)$. Each family's contribution is scaled by $(0.15, 0.25, 0.35, 0.25)$ before averaging across the batch. Programs that fail to parse or crash receive no score and are not registered. Evolution uses five islands, three independent completions from the model per sampled database prompt, cluster sampling with a temperature schedule, periodic island resets, and by default a one-hour wall-clock budget; pending requests are allowed to finish before shutdown. Candidates execute in a restricted NumPy environment with injected FLOP and reward hooks.

The retained program (Table 13) is still a classical stationary iteration expressed entirely through dense `numpy` kernels

```
@funsearch.evolve
def iterative_solver(A, b, x0, max_iter=10, tol=1e-6):
    """Iterative solver to evolve. Returns (solution, convergence_history, iterations)."""
    x = x0.copy()
    convergence_history = [x.copy()]
    D = np.diag(np.diag(A))
    R = A - D

    for i in range(max_iter):
        x_new = np.linalg.solve(D, b - R @ x)
        convergence_history.append(x_new.copy())
        if np.linalg.norm(x_new - x) < tol:
            return x_new, convergence_history, i + 1
        x = x_new

    return x, convergence_history, max_iter
```

*Specification docstring (prepended to the program template): OpenAI system message:*

```
You are a Python code completion assistant. The last function in the code has an empty body.
Write a complete function body (4 spaces indented) that improves upon the previous versions.
Only return the function body code, no explanations or function signature.
```

*OpenAI user message prefix; the implementation appends the full database prompt after the final blank line:*

```
Make only minor, local changes to the previous versions. Be terse and concise:
```

*Table 12.* **FunSearch baseline:** initial evolvable routine (only the body is later rewritten by the LLM), template-level instruction embedded in the serialized specification, and fixed OpenAI message prefixes concatenated before the database prompt.

(solve, lstsq, matmuls); it does not introduce new compositional structure such as sketching, leverage-aware sampling, or problem-specific preconditioning. Mutation budget is spent on rearranging black-box library calls and local robustness, not on expanding the algorithm class that RL4RLA searches via explicit operators.

This baseline therefore explores Python inside one iterative template under strong LLM inductive bias, whereas RL4RLA searches over typed operator programs. Figure 7 reports the empirical comparison.

```
def iterative_solver(A, b, x0, max_T=10, tol=1e-14):
    x = x0.copy().astype(float)
    b = np.asarray(b).astype(float).ravel()
    convergence_history = [x.copy()]
    start_time = time.perf_counter()

    L = np.tril(A)
    U = A - L

    for i in range(max_T):
        rhs = b - U @ x
        try:
            x_new = np.linalg.solve(L, rhs)
        except np.linalg.LinAlgError:
            x_new, *_ = np.linalg.lstsq(L, rhs, rcond=None)
        x_new = x_new.astype(float)
        convergence_history.append(x_new.copy())
        if np.linalg.norm(A @ x_new - b) < tol:
            end_time = time.perf_counter()
            return x_new, convergence_history, end_time - start_time
        x = x_new
    end_time = time.perf_counter()
    return x, convergence_history, end_time - start_time
```

*Table 13.* **FunSearch outcome (1-hour run):** best retained iterative routine used for baseline plots. Aside from bookkeeping, each step is a dense matrix–vector multiply followed by a triangular solve or least-squares fallback from off-the-shelf kernels.

