# OpenReview forum: "RL4RLA: Teaching ML to Discover Randomized Linear Algebra Algorithms Through Curriculum Design and Graph-Based Search"
_ICML.cc/2026/Conference — ICML 2026 regular_

### Official Review · Reviewer_y4kh · 2026-03-10

**Soundness:** 3
**Presentation:** 3
**Significance:** 3
**Originality:** 3
**Overall Recommendation:** 4
**Confidence:** 3

**Summary:**

This paper proposes RL4RLA, a framework for automatically discovering randomized linear algebra algorithms by searching over symbolic programs built from a typed library of 17 fixed linear algebra operators. Its main contributions are a curriculum over problem instances that decomposes algorithm discovery into staged refinements, and a Monte Carlo Graph Search procedure that merges equivalent partial programs to reduce redundant exploration. They are able to rediscover known algorithms from RLA.

**Compliance With Llm Reviewing Policy:**

Affirmed.

**Final Justification:**

The authors have adequately addressed my main concerns. While the scope of the setting remains somewhat narrow, the paper presents an interesting idea for exploring algorithms in RLA. I therefore recommend a weak accept.

**Key Questions For Authors:**

- The authors mention “delayed and expensive rewards” around line 143; could they quantify this more explicitly?
- Did the method discover any genuinely novel algorithm for the target problem? Such a result would strengthen the paper considerably. If not, could the authors clarify whether this is feasible in principle and what concrete steps would be needed to move in that direction?
- Could the authors clarify how the curriculum was designed? More specifically, how should one construct such a curriculum for other RLA problems?
- Could the authors comment on the generality of the approach and its potential applicability beyond the least-squares / linear-system settings studied here?
- A related line of work searches directly over program space using LLMs, with the model serving both as a mutation operator and as a source of inductive bias (e.g., AlphaEvolve). Are the authors aware of this direction, and could they discuss how it relates to their approach?
- Could the authors include additional benchmarks on non-synthetic data?
- In Definition 4.1, the authors define a curriculum stage as a tuple consisting of a “fixed linear system.” Does this mean that search is performed on only a single instance per stage?

**Limitations:**

yes

**Strengths And Weaknesses:**

**Strengths:**
- RLA is a highly relevant field, and the combination with RL is especially compelling.
- The pipeline successfully rediscovers known algorithms from RLA.
- The ablations show that the curriculum design is relevant.
- Figure 1 illustrates the framework nicely.
- Section 3 makes the manuscript quite accessible.
- Section 4 describes the methodology in sufficient detail to enable replication of the algorithms.

**Weaknesses:**
- The authors state that “each curriculum stage is associated with a single algorithm,” which suggests that substantial effort may be required to design the curriculum and may hinder scalability.
- The paper is framed as a general framework for RLA algorithm discovery, but the experiments are mostly limited to least-squares and linear-system settings. As a result, the evidence supports effectiveness on a narrow subfamily of RLA problems, but not yet broad generality across randomized low-rank approximation, SVD, or other core RLA tasks.
- The experiments rely on synthetic results, despite the authors’ mention of RLA’s “essential role in scientific computing and machine learning.”
- The related work section requires significant expansion around automated algorithm discovery, especially around LLM-driven program discovery such as AlphaEvolve (https://arxiv.org/pdf/2506.13131) and ShinkaEvolve (https://arxiv.org/abs/2509.19349).

**Minor:**
- Section 5 should be “Experiment**s**”.

My main concerns are the relatively narrow experimental scope and the corresponding strength of the paper’s claims of generality. Overall, the paper is a promising proof of concept for rediscovering known RLA algorithms, but the current evidence is still limited to a fairly specific synthetic least-squares setting and does not yet demonstrate broad applicability across RLA or clear discovery of genuinely novel algorithms.

---

> ### Author Rebuttal · Authors · 2026-03-31
>
> We thank the reviewer for their detailed and constructive feedback. Supporting material is available at [link](https://anonymous.4open.science/r/ICML-RL4RLA-BCEC/README.md).
>
> ---
>
> **Q1. Quantify the delayed and expensive rewards in MCTS**
>
> Rewards are delayed because feedback is only available after executing a complete candidate program for $T$ iterations, so partial programs provide no intermediate signal. Each rollout requires full numerical execution, with cost ranging from $O(T\cdot \text{nnz}(A))$ to $O(T\cdot mn)$ depending on operators. These factors make naive exploration intractable and directly motivate curriculum design and MCGS.
>
> |program depth|MCTS avg playout|relative cost|
> |--|--|--|
> |1-step|2.12 ms|1×|
> |2-step|82.12 ms|~39×|
> |3-step|143.13 ms|~68×|
>
> ---
>
> **Q2. Discovery of novel algorithms**
>
> We do not claim discovery of a broadly valid new algorithm. However, RL4RLA can in principle, discover novel compositions. In exploratory runs, we found unconventional programs that worked only in restricted regimes, such as square PSD systems, and therefore did not present them as new algorithmic contributions.
>
> Moving from rediscovery to novel algorithm discovery would require: (i) larger search budgets for deeper exploration and broader operator libraries, and (ii) efficient post-search validation. We will clarify this point in the revision.
>
> ---
> **W1.& Q3. Constructing a curriculum**
> Our curriculum is designed through progressive failure-mode decomposition where we begin with a simple solvable instance, search to obtain a naive baseline algorithm, evaluate its main failure mode on harder instances, then construct the next stage to isolate and highlight that failure mode. This process is repeated until the discovered algorithm performs competitive on target evaluation. For other RLA problems, the same principle applies as demonstrated through an example on spectral problem in the next question.
>
> ---
> **W2.& Q4. Generality of the approach and its potential applicability beyond the least-squares / linear-system settings studied?**
>
> We agree that the current experiments emphasize least squares and linear system settings. However, RL4RLA is built around problem-agnostic components such as explicit symbolic programs, curriculum-guided search, and MCGS, while adaptation to a new problem mainly enters through the operator set and reward design.
>
> We instantiated RL4RLA on a symmetric PSD eigenvalue problem with one added primitive (`VEC_NORMALIZE`) and a Rayleigh quotient reward, leaving all other components unchanged. Across 5 seeds, all three curriculum stages succeeded, rediscovering power iteration (C0, C1) and sketched power iteration (C2) (see Figure 1).
>
>
> ---
> **W4.& Q5. LLM-driven program search**
>
> LLM-driven approaches use the model as both mutation operator and inductive bias, which biases search toward training distribution and risks local optima. RL4RLA avoids pretrained priors, allowing exploration beyond common code patterns.
>
> We also compare with AlgoTune [1] and FunSearch [2]. These baselines often rely on library solvers or dense routines rather than discovering new RLA algorithms. FunSearch candidates scale poorly across diverse linear systems, with higher runtime and weaker residuals. We will include these comparisons in the revision.
>
>
> ---
> **W3.& Q6. Experiments on non-synthetic data**
>
> Our discovery procedure uses synthetic instances by design, since the goal is algorithm discovery rather than dataset-specific fitting. Synthetic systems let us control conditioning, dimensionality, and leverage structure, which isolates the failure modes that drive curriculum construction. The symbolic program representation then makes the discovered algorithms directly verifiable.
>
> To complement this controlled setting, we also evaluate the discovered algorithms on the real-world YearPredictionMSD dataset. These results provide initial evidence that the learned algorithmic structures transfer beyond the synthetic systems used during search.
>
> |Metric| Precond GD |Sketched Precond GD|Precond Weighted SGD|Block RK|
> |:--|--:|--:|--:|--:|
> |Residual (↓)|8.69e-01|8.69e-01|9.13e-01|9.19e-01|
> |Time(s) (↓)|85.64|103.46|92.22|159.14|
>
> ---
> **Q7. Does "fixed linear system" mean only a single instance per stage?**
>
> No, each curriculum stage defines a family of linear systems parameterized by relevant numerical properties (e.g., condition number range, leverage score distribution). During search, each program evaluation samples a fresh random instance from this family, ensuring discovered algorithms generalize across the family rather than overfit to a single matrix.
>
> ---
> We hope our response has addressed your concerns. Should any questions remain, we are happy to discuss further, and we hope you will consider revising your assessment.
>
> [1] AlgoTune: Can Language Models Speed Up General-Purpose Numerical Programs? NeurIPS 2025
> [2] Mathematical discoveries from program search with large language models, Nature, 2023

---

> > ### Author Rebuttal · Reviewer_y4kh · 2026-04-02
> >
> > I thank the authors for their rebuttal. My concerns have been adequately addressed. I therefore raise my score to weak accept.

---

> > > ### Author Response · Authors · 2026-04-02
> > >
> > > We sincerely thank the reviewer for their acknowledgment and for kindly raising their score.

---

### Official Review · Reviewer_pieK · 2026-03-11

**Soundness:** 3
**Presentation:** 3
**Significance:** 3
**Originality:** 3
**Overall Recommendation:** 4
**Confidence:** 4

**Summary:**

This paper formulates automated discovery of randomized linear algebra algorithms as a reinforcement-learning problem over explicit symbolic programs built from typed linear-algebra primitives. Each candidate method has a setup stage and an iteration stage, and legality constraints together with dead-code elimination ensure that the search stays within executable, canonical programs. The main methodological idea is to pair this symbolic program search with a numerical curriculum: each stage introduces one new failure mode in a constructed linear system and asks the search process to augment the current algorithm with one additional component.
The search procedure combines curriculum-guided discovery, Monte Carlo Graph Search (MCGS), a weighted reward over solution accuracy, convergence decay, computational complexity, and conditioning, and LUCB-based adaptive stopping. In experiments, the framework rediscovered Landweber iteration, least-squares gradient descent, preconditioned and sketched preconditioned gradient descent, subsampled and weighted variants, Block Randomized Kaczmarz, and Newton Sketch. The empirical evidence shows that graph-based search reduces playouts relative to tree-based MCTS, especially on more compositional targets, and that the full staged curriculum is necessary for Newton Sketch, where no-curriculum and partial-curriculum variants fail completely. Overall, the authors focus on the theme of using reinforcement learning as a structured search procedure for discovering explicit, interpretable RLA algorithms rather than opaque learned optimizers.

**Compliance With Llm Reviewing Policy:**

Affirmed.

**Final Justification:**

I recommend the paper as borderline accept.

**Key Questions For Authors:**

1. How sensitive are the reported discoveries to the specific curriculum construction, including the chosen fixed stage environments and the stage-wise reward weights? A robustness study across alternative stage designs would materially strengthen my confidence in the generality of the method.

2. How exactly is “semantic equivalence” established after automated program comparison? Please clarify the canonicalization pipeline, the role of manual validation, and whether these checks can be released to support reproducibility.

3. Can the discovered algorithms be evaluated on substantially different unseen matrix families without changing the reward design or search hyperparameters? Positive evidence here would raise my assessment of significance.

4. Why is the baseline set limited to MCTS and MCGS variants? A modest comparison to another program-search or evolutionary baseline would help clarify how much of the gain is specific to graph reuse and curriculum design.

5. For the Newton Sketch results, please spell out the precise meanings of curriculum stages C0–C3 and which intermediate capability makes the full curriculum uniquely successful. A clearer explanation would further support the reachability claim.

**Limitations:**

Partially. The impact statement appropriately discusses interpretability, the need for validation before deployment, and the computational cost of search. However, the paper should more explicitly acknowledge three methodological limitations: dependence on synthetic curriculum design, dependence on hand-chosen reward weights, and the gap between rediscovery on constructed environments and generalization to broader numerical settings. A brief dedicated limitations paragraph in the main text would improve balance.

**Strengths And Weaknesses:**

Soundness. The core RL formulation is coherent: states are partial programs, actions insert typed operations, terminal programs are executed and scored, MCGS merges equivalent partial algorithms, and LUCB avoids arbitrary stopping rules. The strongest evidence is not just the end-to-end success table, but the stage-wise and ablation results. The paper reports that curriculum completion success is about 75–80% across the linear-system families, graph-based search reduces total playouts by roughly 2×–3× relative to MCTS, and the full Newton Sketch curriculum achieves 100% success while no-curriculum and partial-curriculum variants fail. Within the constructed environments, these results support the central claims well. My main reservation is external validity: each curriculum stage is built around carefully designed synthetic systems and stage-specific reward weights, so the evidence mainly demonstrates successful rediscovery under those designed conditions rather than broad transfer. A second limitation is that the comparison set is narrow for an RL-focused paper; most comparisons are among MCTS and MCGS variants, so the reader does not learn how the approach fares against stronger alternative program-search baselines.

Presentation. The paper is generally clear and well structured. The flow from preliminaries to symbolic search to curriculum design to experiments is easy to follow, and Table 1 plus Figures 1–4 communicate the intended search dynamics effectively. The explicit program representation is a notable presentation strength because it makes the output interpretable. That said, several details deserve sharper explanation in a revision: the semantic-equivalence validation procedure is only briefly described, the amount of manual validation is not fully quantified, and the paper would benefit from a more direct discussion of sensitivity to the chosen curriculum stages, reward weights, and fixed stage environments. There are also minor proofreading issues and small notation inconsistencies that should be cleaned up.
Significance. The problem addressed is important and well matched to RL-style search. The paper argues convincingly that RLA has modular, compositional structure and that this makes symbolic discovery feasible if the environment is designed carefully. I view the contribution as meaningful because it offers a concrete recipe for making RL practical in a hard algorithm-discovery domain: encode domain knowledge through failure-mode curricula, search explicit programs, and reuse search effort through graph merging. The current significance is moderated by the synthetic setting and the emphasis on rediscovery rather than unexpectedly new algorithms, but the framework is likely to be useful for future work in automated numerical-method discovery.

Originality. The contribution is original primarily in its integration and framing rather than in proposing a wholly new RL primitive. The combination of curriculum stages organized around numerical failure modes with graph-based symbolic search over typed linear-algebra programs is creative and well motivated. The work also shows that the same framework can recover multiple classical RLA paradigms under different curricula. My view is that the originality claim is solid, but should remain calibrated: the paper demonstrates a strong and novel system-level synthesis more than a fundamentally new theory of RL or RLA discovery.

Overall assessment. I find the submission technically solid and interesting, with its strongest case on the RL design side. My main concern is that the paper shows convincingly that carefully engineered curricula make known algorithm families reachable, but not yet that the framework generalizes broadly or discovers genuinely surprising new algorithms beyond the designed search regime.

---

> ### Author Rebuttal · Authors · 2026-03-31
>
> We thank the reviewer for their detailed and constructive feedback. Supporting material is available at [link](https://anonymous.4open.science/r/ICML-RL4RLA-BCEC/README.md).
>
> ---
> **W1. Curricula built around carefully designed systems**
> We agree that the current evidence is strongest under controlled synthetic curricula and will clarify this in the revision. The synthetic setting is intentional, as the goal is algorithm discovery under controlled numerical structure rather than dataset-specific fitting. By varying conditioning, spectral decay, and leverage-score heterogeneity, the curriculum isolates key numerical mechanisms, while the symbolic program representation makes the resulting algorithms directly inspectable and verifiable.
>
> We further provide initial evidence of transfer on the real-world YearPredictionMSD dataset and show that the same framework extends to a spectral problem with minimal changes. These results suggest transfer across both data regimes and problem classes, though broader validation remains future work.
>
> |Metric| Precond GD |Sketched Precond GD|Precond Weighted SGD|Block RK|
> |:--|--:|--:|--:|--:|
> |Residual (↓)|8.69e-01|8.69e-01|9.13e-01|9.19e-01|
> |Time(s) (↓)|85.64|103.46|92.22|159.14|
>
> ---
> **W2. & Q4. Comparison with Program-search baselines**
>
> The current baselines focus on MCTS and MCGS to evaluate curriculum-guided search and graph reuse under a fixed program space. We agree that broader program search baselines would strengthen the paper, and thus include LLM-based approaches such as AlgoTune [1] and FunSearch [2] in the supplementary.
>
> LLM baselines often rely on existing library solvers rather than discovering new RLA algorithms. FunSearch candidates also scale poorly across diverse linear systems, with orders-of-magnitude higher runtime and noncompetitive residuals on both well-conditioned and ill-conditioned systems. We will add these comparisons in the revision.
>
> ---
> **W3. & Q2. Semantic-equivalence validation procedure**
>
> The semantic equivalence check proceeds in two steps: (i) automated symbolic canonicalization using algebraic simplification rules (e.g., collapsing A^T A products, eliminating identity operators, normalizing scalar multiplications), followed by (ii) execution-based equivalence testing on a held-out set of random matrix instances. Two programs are declared equivalent if and only if both steps agree. Manual validation is limited to spot-checking declared-equivalent pairs. We will release the canonicalization pipeline as part of our code release to support reproducibility.
>
> ---
> **W4. & Q1. Sensitivity to curriculum stages and reward weights not directly addressed**
>
> We thank the reviewer for their suggestion. However, the reward weights are not arbitrary hyperparameters in our framework. Each reward component corresponds to a numerical objective, such as accuracy, convergence, cost, or stability. The weights are adjusted across curriculum stages so that the reward aligns with the dominant failure mode highlighted at that stage. In this sense, the weights mainly control the relative emphasis among objectives, while the broader curriculum design governs the progression of algorithmic structure.
>
> ---
> **Q3. Can the discovered algorithms be evaluated on substantially different unseen matrix families without changing the reward design or search hyperparameters?**
>
> For a fixed discovered algorithm, evaluation on unseen matrix families does not require changing the reward design or search hyperparameters, since those only affect the discovery phase and not execution.
>
> If the question instead concerns advancing to a new curriculum stage that introduces a new matrix family, then some change in reward weighting is by design. Our curriculum is constructed through failure mode analysis, and the reward is adjusted to emphasize the newly exposed objective at that stage.
>
> ---
> **Q5. Clarifications on curriculum stages C0–C3**
>
> The table illustrates each stage:
>
> | Stage | Algorithm | Update Terms |
> |---|---|---|
> | C0 | Logistic forward pass | `SIGMOID` + register structure |
> | C1 | Logistic GD | residual calculation |
> | C2 | Full Newton | `DIAG_MATRIX` → `BUILD_WEIGHTED_MATRIX` → `COMPUTE_INVERSE_HESSIAN` |
> | C3 | Newton Sketch | insert `SKETCH` before `COMPUTE_INVERSE_HESSIAN` |
>
> Specifically, C3 is constructed by replacing the exact Hessian with $\hat{H}_t = (SD_t^{1/2}A)^\top(SD_t^{1/2}A)$. This is only discoverable if C2's Hessian construction pattern already exists. This stage reduces computational complexity while maintaining accuracy comparable to the previous stage (C2).
>
> ---
> We hope our response has addressed your concerns. Should any questions remain, we are happy to discuss further, and we hope you will consider revising your assessment.
>
> [1] AlgoTune: Can Language Models Speed Up General-Purpose Numerical Programs? NeurIPS 2025
> [2] Mathematical discoveries from program search with large language models, Nature, 2023

---

> > ### Author Rebuttal · Reviewer_pieK · 2026-04-03
> >
> > I would like to thank the authors for their responses. I will keep my positive score.

---

> > > ### Author Response · Authors · 2026-04-03
> > >
> > > We sincerely thank the reviewer for acknowledging our work and for maintaining their positive assessment.

---

### Official Review · Reviewer_kYWc · 2026-03-12

**Soundness:** 2
**Presentation:** 3
**Significance:** 2
**Originality:** 2
**Overall Recommendation:** 4
**Confidence:** 4

**Summary:**

This paper presents RL4RLA, a general reinforcement learning (RL) framework for automated discovery of interpretable, symbolic randomized linear algebra (RLA) algorithms. Overall, a fundamental problem presented by the study is that standard RL approaches fail to efficiently discover high-performance RLA algorithms, due to the inherent sparse reward landscapes and vast combinatorial search spaces in this domain. To tackle this challenge, the authors propose two core technical designs: a numerical curriculum that progressively increases problem difficulty to encode RLA-specific inductive biases, and Monte Carlo Graph Search (MCGS) that eliminates redundant exploration by merging equivalent partial algorithm states. Overall, the authors focus on th theme of building the first general-purpose framework for automated RLA algorithm discovery, which bridges the gap between the rapid progress of automated algorithm design in other domains and the under-explored RLA field. The authors demonstrate that RL4RLA can reliably rediscover multiple state-of-the-art RLA methods (including sketch-and-precondition solvers, Randomized Kaczmarz, and Newton Sketch), and can generate algorithm variants with customized trade-offs between accuracy, speed, and numerical stability.

**Compliance With Llm Reviewing Policy:**

Affirmed.

**Final Justification:**

My main concerns have been largely clarified, but this paper still requires considerable revision. Therefore, I will maintain the "barely acceptable" rating.

**Key Questions For Authors:**

1. The experiments are primarily based on synthetic linear systems with controlled numerical properties. Have you evaluated the performance of RL4RLA-discovered algorithms on real-world large-scale linear algebra tasks (e.g., real regression datasets, PDE discretization systems)?

2. You mention extending the framework to broader numerical domains in future work. What are the key challenges you anticipate in adapting the current curriculum design and operator library to non-least-squares tasks (e.g., eigenvalue problems)?

3. The reward function balances multiple metrics including accuracy, convergence, computational cost, and numerical stability. How sensitive is the structure of discovered algorithms to changes in the reward weights? Addressing this sensitivity analysis will improve confidence in the flexibility of your framework.

4. You show that MCGS greatly reduces redundant exploration compared to standard MCTS. Could you provide additional analysis on how the state merging mechanism in MCGS scales with increasing program length and operator library size? This will help readers understand the limits of the proposed search method for more complex numerical algorithms.

**Limitations:**

yes

**Strengths And Weaknesses:**

## Strengths

- The paper is well-structured with a clear and logical narrative. It effectively motivates the problem, explains the methodology step-by-step, and presents results in an accessible manner. The curriculum progression example and framework overview greatly help readers understand the core approach.
-  This work fills a critical gap in the literature: while automated algorithm discovery has achieved great success in optimization, matrix multiplication, and sorting, there is no general-purpose framework for RLA algorithm discovery prior to this work. Given that RLA is a cornerstone of large-scale scientific computing and machine learning, this framework has broad potential impact for both academic research and practical applications.
-  Although curriculum learning and MCGS are existing techniques, this work provides a novel, domain-specific combination of these methods tailored for RLA algorithm discovery. It uniquely formulates RLA algorithm design as a symbolic program search over linear algebra primitives, ensuring the outputs are fully interpretable and implementable, which clearly distinguishes it from prior black-box learned RLA methods.

## Weaknesses
-  The experiments are mainly conducted on synthetic linear systems, and the paper lacks detailed evaluation of the generalization ability of discovered algorithms on real-world large-scale linear algebra tasks.
-  There is a typographical error in line 154, where "premitives" is misspelled (the correct spelling is "primitives").
-  The current framework is limited to linear least-squares related tasks, and the paper does not provide preliminary evidence of its extensibility to broader numerical linear algebra problems mentioned in the conclusion (e.g., eigenvalue problems, PDE solvers).
-  The core technical components are adapted from existing work, and the paper could better elaborate on the domain-specific modifications made to curriculum learning and MCGS for the RLA discovery task, to further highlight its novelty.

---

> ### Author Rebuttal · Authors · 2026-03-31
>
> We thank reviewer for their detailed and constructive feedback. Supporting material is available at [link](https://anonymous.4open.science/r/ICML-RL4RLA-BCEC/README.md).
>
> ---
> **W1. & Q1. Experiments entirely on synthetic linear systems**
>
> Our discovery is conducted on synthetic instances by design, since the goal is algorithm discovery rather than dataset specific fitting. Synthetic systems let us control key numerical properties such as conditioning, dimensionality, and leverage structure, and therefore isolate the failure modes that drive curriculum construction. Combined with the symbolic program representation, this makes the discovered algorithms directly inspectable and allows their correctness and numerical behavior to be analyzed independently of any particular dataset.
>
> To complement this controlled discovery setting, we also evaluate the discovered algorithms on the real world YearPredictionMSD dataset. These results provide initial evidence that the learned algorithmic structures transfer beyond the synthetic instances used during search.
>
> |Metric|Precond GD|Sketched Precond GD|Precond Weighted SGD|Block RK|
> |:-|--:|--:|--:|--:|
> |Residual (↓)|8.69e-01|8.69e-01|9.13e-01|9.19e-01|
> |Time (s) (↓)|85.64|103.46|92.22 |159.14|
>
> ---
> **W2. Typographical error**
>
> We thank the reviewer for their careful reading. We will fix the typo in the revised manuscript.
>
> ---
> **W3. & Q2 Adapting RL4RLA to Spectral Problems and challenges**
>
> We agree that the current experiments emphasize least squares and linear system settings. However, RL4RLA is built on problem-agnostic components such as explicit symbolic programs, curriculum guided search, and MCGS, while adaptation to a new problem mainly enters through the operator library and reward design.
>
> We instantiated RL4RLA on a symmetric PSD eigenvalue problem. The search procedure and curriculum logic stays unchanged. The only additions were one normalization primitive, `VEC_NORMALIZE`, and a Rayleigh quotient based reward. Across 5 seeds, all three curriculum stages succeeded, rediscovering power iteration in C0 and C1 and sketched power iteration in C2. We will add this result in the revision to better demonstrate applicability beyond least squares settings.
>
> ---
> **W4. Domain-specific Modifications**
>
> We agree that curriculum learning and MCGS exist as general methods. Our contribution lies in two non-trivial adaptations required for symbolic RLA discovery. First, state merging in MCGS requires recognizing semantically equivalent programs produced by different action orderings, which is non-trivial and has no direct analogue in prior MCGS applications. Second, unlike standard curricula where difficulty is a scalar, our curriculum is constructed by identifying key matrix property (conditioning, leverage, dimensionality) that exposes failure patterns, whose necessity is supported by Table 6: all partial curriculums for Newton Sketch fails at 0% success regardless of budget, while the full curriculum achieves 100%. We will make this separation more explicit in the revision.
>
> ---
> **Q3. Sensitivity analysis on reward weights**
>
> We thank the reviewer for their suggestion. However, the reward weights are not arbitrary hyperparameters in our framework. Each reward component corresponds to a numerical objective, such as accuracy, convergence, cost, or stability. The weights are adjusted across curriculum stages so that the reward aligns with the dominant failure mode highlighted at that stage. In this sense, the weights mainly control the relative emphasis among objectives, while the broader curriculum design governs the progression of algorithmic structure.
>
> ---
> **Q4. How MCGS state merging scale with problem size**
>
> To analyze scalability, we ran MCGS and MCTS on a controlled search task (rediscovering `sketched_precond_gd`, a base program of 8 ops) across varying operator library sizes (17, 20, 25) and target program lengths (dist 8, 10, 12), with 20 seeds each.
>
> |Library size \ Program length|8|10|12|
> |-:|--:|--:|--:|
> |17|0.578|0.530|0.520|
> |20|0.578|0.527| — |
> |25|0.533|0.499| — |
>
> *Revisit ratio = fraction of search steps redirected to already-visited states. MCTS = 0.000 in all settings.*
>
> MCGS consistently achieves revisit ratios of 0.50–0.58, meaning over half of all search steps exploit previously merged states rather than spawning redundant subtrees. Two trends are visible: revisit ratio decreases modestly as program length grows (0.578→0.520, lib=17) and as library size grows (0.578→0.533, dist=8), since both increase the structural diversity of partial programs. “Importantly, both effects are moderate, so the merging benefit degrades gracefully rather than collapsing, suggesting that MCGS remains effective as search complexity increases.
>
> ---
> We hope our response has addressed your concerns. Should any questions remain, we are happy to discuss further, and we hope you will consider revising your assessment.

---

> > ### Author Rebuttal · Reviewer_kYWc · 2026-04-01
> >
> > I would like to thank the authors for the additional experiments and the detailed response, which have addressed my concerns. I will maintain my score of 4: Weak Accept, reflecting my positive assessment of the work.

---

> > > ### Author Response · Authors · 2026-04-01
> > >
> > > We sincerely thank the reviewer for acknowledging our work and for maintaining their positive assessment.

---

### Official Review · Reviewer_fpvd · 2026-03-13

**Soundness:** 2
**Presentation:** 2
**Significance:** 3
**Originality:** 3
**Overall Recommendation:** 4
**Confidence:** 2

**Summary:**

This work proposes a reinforcement-learning framework for automatically discovering randomized linear algebra (RLA) algorithms. It frames the algorithm discovery as a Directed Acyclic Graph (DAG)-like MDP where actions consist of inserting operations into a partial symbolic program. To efficiently traverse this DAG, Monte Carlo Graph Search (MCGS) is employed with curriculum design. Experiments show that the framework can rediscover well-known algorithms while improving search efficiency compared with standard MCTS

**Compliance With Llm Reviewing Policy:**

Affirmed.

**Final Justification:**

My main concern is substantially clarified, but not fully resolved, especially regarding the task-specific curriculum design. I will therefore keep my score at weak accept.

**Key Questions For Authors:**

- how can the framework be used to discover genuinely new algorithms rather than rediscovering existing ones?

**Limitations:**

yes

**Strengths And Weaknesses:**

### Strengths
- **Interpretable symbolic programs:** the proposed method generates algorithms as explicit symbolic programs, so it ensures the results are human-readable and mathematically verifiable
- **Verification via rediscovery:** The framework successfully rediscovers several classical RLA algorithms, which is good evidence that the search space and optimization procedure are meaningful.
- **More efficient search through graph-based merging:** experiments clearly show how MCGS improve search efficiency by avoiding redundant exploration compared to the standard MCTS

### Weaknesses
- **Strong reliance on heuristics:** Performance seems heavily dependent on manually designed curricula and reward weights, which may limit robustness and scalability.
- **Domain-Specific components**: Although the search procedure is general, many key components—such as the operator grammar, legality constraints, curriculum design, and reward function—are heavily domain-specific to randomized linear algebra.

---

> ### Author Rebuttal · Authors · 2026-03-31
>
> We thank reviewer for their detailed and constructive feedback. Supporting material is available at [link](https://anonymous.4open.science/r/ICML-RL4RLA-BCEC/README.md).
>
> ---
> **W1. Strong reliance on manually designed curricula and reward shaping**
>
> We agree that curriculum and reward design play an important role, but they follow a systematic and principled recipe rather than ad hoc tuning. Our curriculum is constructed through failure mode analysis. Starting from a simple solvable instance, we identify the dominant failure mode on harder instances and design the next stage to isolate it. The reward weights are then adjusted to emphasize the newly exposed objective at each stage—such as accuracy early on, and conditioning or complexity later. This adjustment is not arbitrary; it is a direct consequence of which failure mode the stage is designed to expose. The reward weights change because the optimization target changes rather than from ad hoc tuning.
>
> More importantly, this recipe transfers across problem domains. We instantiated RL4RLA on a symmetric PSD eigenvalue problem using the same search procedure and curriculum logic, adding only one normalization primitive `VEC_NORMALIZE` and a Rayleigh quotient-based reward. Across 5 seeds, all three curriculum stages succeeded in rediscovering power iteration (C0, C1) and sketched power iteration (C2). We will include this result in the revision to demonstrate that the framework's structure generalizes beyond specific problem settings, i.e., least squares.
>
> ---
> **W2. Domain-specific components**
>
> We thank the reviewer for raising this point. The key distinction is that the search procedure and MCGS are domain-agnostic. What requires domain knowledge is the interface layer, including the operator grammar, legality constraints, curriculum stages, and reward definition. This interface is small and requires modest problem-specific adaptation. As the eigenvalue experiment above illustrates, adapting RL4RLA to a new problem class required specifying one additional primitive and one reward function while the rest of the framework transferred unchanged. We view this interface as a feature rather than a limitation, as it ensures that discovered algorithms are interpretable, executable, and grounded in the target domain's numerical structure. We will clarify this decomposition in the revision.
>
> ---
> **Q1. How can the framework discover genuinely new algorithms?**
>
> This is an important question, and we want to be precise about what RL4RLA can currently claim. The search space is defined by the operator grammar, and any algorithm discoverable by the framework must be expressible as a composition of those primitives. Within that space, genuinely novel combinations can emerge since they are not constrained to be known algorithms. In exploratory runs, we have observed programs that do not correspond to named methods in the literature. However, upon closer inspection, these turned out to be functioning only within the narrow regime (e.g., square PSD systems) and not broadly applicable. Therefore, we did not present the result as a new algorithmic contribution. Nevertheless, it demonstrates the feasibility of discovering novel algorithms in principle.
>
> Moving from rediscovery to validated novel discovery would require: (i) broader operator libraries and larger search budgets for deeper exploration, and (ii) systematic post-search validation with formal analysis. We will include these findings and clarify this distinction in the revision.
>
> ---
> We hope our response has addressed your concerns. Should any questions remain, we are happy to discuss further, and we hope you will consider revising your assessment.

---

> > ### Author Rebuttal · Reviewer_fpvd · 2026-04-03
> >
> > My main concern is substantially clarified, but not fully resolved. I agree that the curriculum and reward design are not purely ad hoc, and the authors did a good job explaining the failure-mode-driven logic behind them. The clarification on novelty is also helpful.
> >
> > However, I still view the framework as strongly dependent on manually designed, task-specific curricula and interface choices. In my view, this remains the main practical bottleneck and also limits how general the method currently is. The rebuttal weakens this concern, but does not remove it. I therefore keep my score at weak accept.

---

> > > ### Author Response · Authors · 2026-04-03
> > >
> > > We thank the reviewer for the thoughtful follow-up and continued feedback. We are glad that our clarification on the failure-mode-driven curriculum design and the novelty claim was helpful.
> > >
> > > On the remaining concern, we would like to clarify that the problem-specific pieces in RL4RLA are a thin interface layer rather than a different workflow for each task. The explicit symbolic program search, failure-mode-driven curriculum, and the MCGS-based graph reuse remain unchanged across problem classes. What changes is limited to the compact task interface needed to make the target problem executable, such as the primitive set and reward definition. In particular, our eigenvalue experiment already shows that moving beyond least squares required only minimal interface changes while leaving the search procedure itself intact.
> > >
> > > Our claim is therefore not that RL4RLA removes all task specification, but that it reduces manual solver design to specifying a minimal executable interface. We will make this decomposition more explicit in the revision so that the paper better separates the general method from the task-specific interface.

---

### Decision · Program_Chairs · 2026-04-30

**Decision:**

Accept (regular)

**Comment:**

This paper makes an important contribution by proposing the first automatic RLA algorithm discovery algorithm by leveraging RL techniques that have been successful for automatic optimization and matrix manipulation algorithms. To achieve this generalization, the authors combined Monte Carlo Graph Search with curriculum learning.

The reviewers agreed that the presentation was strong and the ideas novel. One shortcoming was the heuristics, which limited the strength of the empirical evidence.